# Conditional Flow Matching for Conformal Regression

## Abstract

This paper introduces Conditional flow Matching for conformal Regression (CMR), a novel framework that synergizes simulation-free conditional flow matching with conformal prediction to generate reliable and efficient prediction intervals. Unlike traditional methods that rely on quantile regression or fixed histograms, CMR leverages Continuous Normalizing Flows (CNFs) trained via Conditional Flow Matching (CFM) to accurately model complex, multimodal conditional distributions. To ensure finite-sample coverage guarantees, we introduce a novel nonconformity score defined as the minimum number of generated samples required for the shortest interval to encompass the true outcome. This mechanism allows CMR to dynamically adjust interval widths based on the learned probability density. Extensive experiments on simulated and real-world datasets demonstrate that CMR consistently produces narrower prediction intervals while maintaining the required marginal coverage and achieving superior tail coverage compared to state-of-the-art methods.

## 1 Introduction

In recent years, machine learning and deep learning models (Magdon-Ismail & Atiya, 1998; Meinshausen & Ridgeway, 2006) have achieved remarkable success in regression tasks and beyond. However, while point predictions from these models continue to advance, they often fall short of industrial standards due to unresolved challenges in model trustworthiness and robustness. Conformal prediction (Vovk et al., 2005; Shafer & Vovk, 2008) emerges as a critical framework to bridge this gap: by transforming point estimates into prediction intervals with guaranteed coverage probabilities, it provides statistically valid uncertainty quantification without relying on specific model architectures. Rooted in the principle of data exchangeability, this approach constructs confidence intervals aligned with predefined confidence levels, offering a model-agnostic solution to enhance reliability in real-world deployments.

A key challenge in conformal regression lies in minimizing prediction set sizes while maintaining rigorous coverage guarantees. Most existing conformal methods generally fall into two primary categories: (1) those that derive prediction intervals by leveraging model-based outputs (e.g., decision trees, random forests, or neural networks) to directly estimate interval bounds (Chipman et al., 2010; Papadopoulos et al., 2011; Kivaranovic et al., 2020; Moon et al., 2021); and (2) those that construct prediction sets by inverting conditional density estimates derived from the data (Izbicki et al., 2019; Diamant et al., 2024; Plassier et al., 2024; Zheng & Zhu, 2024). To further enhance performance, numerous approaches incorporate residual-based calibration (Chen et al., 2018b; Lei et al., 2018; Barber et al., 2021) or localized adaptation strategies (Guan, 2023; Colombo, 2024; Cheung et al., 2024; Gil et al., 2024; Hore & Barber, 2025). However, these methods often struggle with complex data distributions (e.g., bimodal distributions), producing excessively large prediction sets that introduce significant uncertainty. Furthermore, certain techniques involve overly intricate procedures, posing practical challenges for implementation. To address these issues, a subset of research (Izbicki et al., 2022; Luo & Zhou, 2025; Gao et al., 2025) has focused on handling data from multiple distributions. Nevertheless, the intervals generated by these

methods are typically not continuous (i.e., unions of disjoint intervals), which significantly limits their applicability in real-world scenarios.

To address the challenges posed by complex distributions, such as excessively wide prediction intervals and operational complexities, we propose a novel and versatile framework: Conditional Flow Matching for Conformal Regression (CMR). By leveraging Conditional Flow Matching, which is theoretically grounded, we learn and generate the target distribution. We then combine this with a new nonconformity score to produce the smallest continuous prediction intervals. Our primary contributions are:

- We introduce a framework that explicitly models the conditional distribution using conditional flow matching and leverages this model to construct prediction intervals with guaranteed coverage.
- We develop a new nonconformity score based on identifying the shortest interval containing the true value among sampled predictions, significantly reducing interval sizes compared to existing methods.
- We demonstrate through rigorous theoretical analysis that our approach provides valid coverage guarantees regardless of model accuracy.
- We empirically validate our method across various simulated and real-world datasets, showing that CMR consistently achieves smaller prediction intervals while maintaining coverage requirements and delivering superior tail coverage performance.
- We decoupled CMR. It outperforms other methods by generating smaller prediction sets. Using Conditional Flow Matching as the base and combining it with other Conformal Prediction approaches yields even smaller sets, with stable conditional coverage and good tail coverage.

## 2 RELATED WORK

### 2.1 CONFORMAL PREDICTION

Conformal prediction (Vovk et al., 2005; Kivaranovic et al., 2020; Romano et al., 2019; Sesia & Candès, 2020; Sesia & Romano, 2021; Wang et al., 2023; Kiyani et al., 2024; Luo & Zhou, 2025; Liu et al., 2025) is a framework for constructing prediction intervals with finite-sample coverage guarantees. Given a desired miscoverage level $\alpha \in (0, 1)$, conformal prediction produces prediction regions that contain the true outcome with probability at least $1 - \alpha$.

Let $\mathcal{D} = \{(X_1, Y_1), \ldots, (X_n, Y_n)\}$ be a dataset of i.i.d. examples from an unknown distribution $P_{XY}$ on $\mathcal{X} \times \mathcal{Y}$. We begin by randomly partitioning the dataset $\mathcal{D}$ into three sets: $\mathcal{I}_{\text{train}}$, $\mathcal{I}_{\text{cal}}$, and $\mathcal{I}_{\text{test}}$. A base predictive model $\widehat{f}(x_i)$ is trained on $\mathcal{I}_{\text{train}}$. Subsequently, we compute a non-conformity score $S_i$ (e.g., absolute residual $|y_i - \widehat{f}(x_i)|$) on the calibration set $\mathcal{I}_{\text{cal}}$. Following the Split Conformal Regression framework (Papadopoulos et al., 2002; Vovk et al., 2005), the $1 - \alpha$ prediction interval for a novel test sample $x_{n+1} \in \mathcal{I}_{\text{test}}$ is constructed as:

$$\mathcal{C}_{n,\alpha}(x_{n+1}) = \{y \in \mathbb{R} : s(x_{n+1}, y) \leq \widehat{q}_{1-\alpha}\}, \tag{1}$$

where $\widehat{q}_{1-\alpha}$ is the $\lceil (1 - \alpha)(|\mathcal{I}_{\text{cal}}| + 1) \rceil / |\mathcal{I}_{\text{cal}}|$-th empirical quantile of the calibration scores. Under the exchangeability assumption, this guarantees:

$$\mathbb{P}(Y_{n+1} \in \mathcal{C}_{n,\alpha}(x_{n+1})) \geq 1 - \alpha. \tag{2}$$

*Conformal Histogram Regression* (Sesia & Romano, 2021) constructs prediction intervals by estimating the full conditional density $\widehat{f}_{Y|X}$ using histograms and finding the shortest interval $(a, b)$ such that:

$$\mathcal{C}^{\text{CHR}}(x_{n+1}) = \arg\min_{a<b}(b - a), \tag{3}$$

$$\text{s.t.} \int_a^b \widehat{f}_{Y|X}(y|x_{n+1}) \, dy \geq 1 - \alpha. \tag{4}$$

## 2.2 Continuous Normalizing Flows

Continuous Normalizing Flows (CNFs) (Chen et al., 2018a; Grathwohl et al., 2018) are a class of generative models that define a probability path from a simple source distribution $p_0$ to a complex target distribution $p_1$ via an Ordinary Differential Equation (ODE). The transformation is defined by:

$$\frac{\mathrm{d}y}{\mathrm{d}t} = v_\theta(t, y, x), \quad y(0) = y_0 \sim p_0, \quad y(1) = y_1 \sim p_1, \tag{5}$$

where $v_\theta(t, y, x)$ is a time-dependent vector field parameterized by a neural network.

## 2.3 Conditional Flow Matching

Conditional Flow Matching (CFM) (Lipman et al., 2023; Tong et al., 2023) is a simulation-free training objective that regresses the neural vector field $v_\theta$ directly to a target vector field $u_t$.

Consider a conditional probability path $p_t(y|y_0, y_1)$ that interpolates between a source sample $y_0 \sim p_0$ and a target sample $y_1 \sim p_1$. If $u_t(y|y_0, y_1)$ is the vector field generating this path, the CFM objective is defined as:

$$\mathcal{L}_{\mathrm{CFM}}(\theta) = \mathbb{E}_{t, p_0(y_0), p_1(y_1), p_t(y|y_0, y_1)} \left[ \left\| v_\theta(t, y, x) - u_t(y|y_0, y_1) \right\|^2 \right]. \tag{6}$$

By minimizing this objective, $v_\theta$ approximates the marginal vector field that transports $p_0$ to $p_1$.

# 3 Conditional Flow Matching for Conformal Regression

In this section, we elaborate on our proposed methodology, **C**onditional flow **M**atching for conformal **R**egression (CMR). Our approach consists of three steps: (1) training a conditional generative model using CFM to estimate $p(y|x)$; (2) calibrating prediction intervals using a novel "shortest-interval" non-conformity score; and (3) constructing valid prediction intervals for new test points.

## 3.1 Learning the Conditional Distribution via CFM

We employ a CNF to model the conditional distribution $p(y|x)$. To ensure efficient training, we adopt the **Conditional Flow Matching (CFM)** formulation.

We define the probability path $p_t$ as a linear interpolation between the source noise $y_0 \sim p_0(y_0) = \mathcal{N}(0, 1)$ and the data $y_1 \sim p(y|x)$, smoothed by a small noise $\sigma$:

$$p_t(y|y_0, y_1) = \mathcal{N}\left(y \,|\, ty_1 + (1-t)y_0, \, \sigma^2\right). \tag{7}$$

The unique vector field $u_t$ that generates this Gaussian probability path is the constant velocity field pointing from $y_0$ to $y_1$:

$$u_t(y|y_0, y_1) = y_1 - y_0. \tag{8}$$

Consequently, the specific regression loss for our model becomes:

$$\mathcal{L}_{\mathrm{CMR}}(\theta) = \mathbb{E}_{\substack{t \sim \mathcal{U}(0,1), \\ x \sim p(x), y_1 \sim p(y|x), \\ y_0 \sim p_0(y_0), \\ y_t \sim p_t(y|y_0, y_1)}} \left[ \left\| v_\theta(t, y_t, x) - (y_1 - y_0) \right\|^2 \right]. \tag{9}$$

This objective allows us to learn the conditional distribution $p(y|x)$ without ODE simulation.

## 3.2 Sampling from the Learned Model

Once trained, the network $v_\theta$ approximates the vector field. To sample from the learned conditional distribution $\widehat{p}(y|x)$, we sample $z \sim \mathcal{N}(0, 1)$ and solve the ODE from $t = 0$ to $t = 1$:

$$\widehat{y} = \mathrm{ODESolve}(v_\theta, z, t \in [0, 1], x). \tag{10}$$

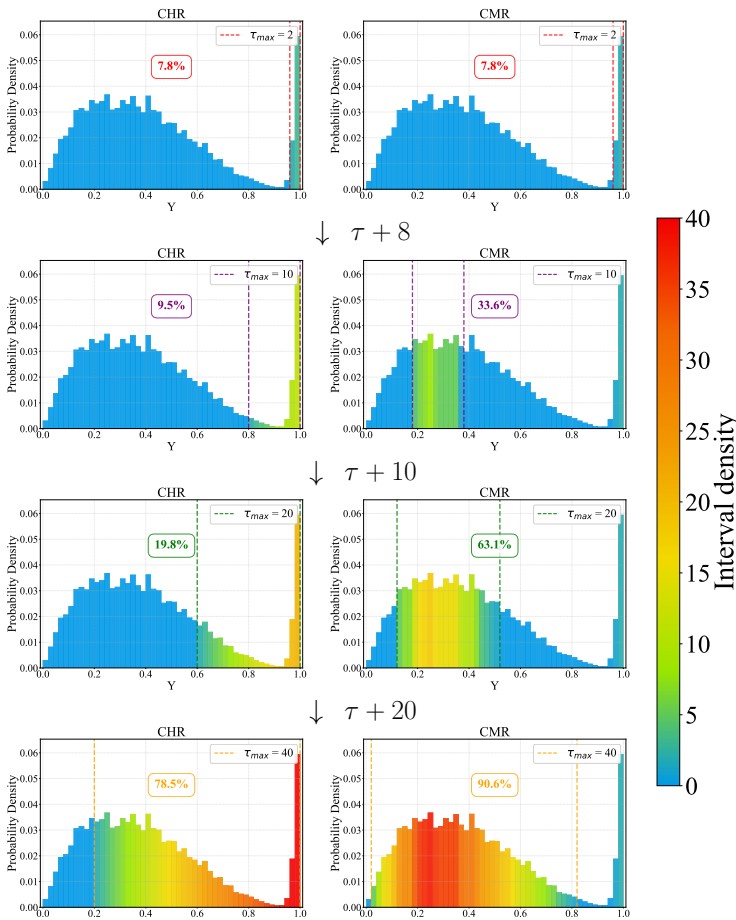

Figure 1: To demonstrate the advantages of our method in handling complex distributions, we visualize the performance of CMR versus CHR in a bimodal setting. The horizontal axis represents the response space, and the vertical-axis denotes the probability density. CMR dynamically selects the contiguous interval that maximizes the total probability density. In contrast, CHR employs an expansion-based strategy, creating a contiguous interval by extending outwards from the highest-density bin. (As shown in the bimodal distribution, if the initial starting bin does not align with the densest aggregate region, CHR requires a significantly larger number of bins to achieve the equivalent probability mass). Consequently, for a fixed number of bins, CMR captures a higher proportion of the total probability mass.

### 3.3 Calibration Procedure

To construct valid prediction intervals, we introduce a non-conformity score based on the *tightness* of the generated distribution. We use a calibration set $\mathcal{I}_{\text{cal}} = \{(x_i, y_i)\}_{i=1}^{|\mathcal{I}_{\text{cal}}|}$. For each calibration point $(x_i, y_i)$:

1. **Generate Samples:** Generate $m$ samples $\{y_i^1, \dots, y_i^m\}$ from the trained model conditioned on $x_i$ via Eq. (10).

2. **Sort:** Sort these samples to obtain the order statistics $\{y_i^{(1)} \leq y_i^{(2)} \leq \dots \leq y_i^{(m)}\}$.

3. **Find Shortest Intervals:** For each possible number of included samples $s \in \{2, \dots, m\}$, we identify the shortest continuous interval that contains exactly $s$ generated samples. Let $W_i(s)$ be this minimum width interval:

$$W_i(s) = [y_i^{(j^*)}, y_i^{(j^*+s-1)}], \quad \text{where } j^* = \arg \min_{1 \leq j \leq m-s+1} (y_i^{(j+s-1)} - y_i^{(j)}). \quad (11)$$

4. **Compute Score:** We define the non-conformity score $L_i$ as the **minimum number of samples** $s$ required for the corresponding shortest interval $W_i(s)$ to cover the true outcome $y_i^{true}$:

$$L_i = \min\{s \in \{2, \ldots, m\} \mid y_i^{true} \in W_i(s)\}. \tag{12}$$

Finally, we compute the quantile threshold $\widehat{q}_{1-\alpha}$ from the calibration scores $\{L_1, \ldots, L_{|\mathcal{I}_{\text{cal}}|}\}$:

$$\widehat{q}_{1-\alpha} = \text{Quantile}_{\lceil(1-\alpha)(|\mathcal{I}_{\text{cal}}|+1)\rceil/|\mathcal{I}_{\text{cal}}|}(\{L_1, \ldots, L_{|\mathcal{I}_{\text{cal}}|}\}). \tag{13}$$

This threshold represents the minimum "sample count" required to ensure coverage.

**Theorem 1** (Marginal Coverage Guarantee). *Let* $(x_i, y_i)$ *be calibration data and* $(x_{n+1}, y_{n+1})$ *be a test point, i.i.d. drawn from* $P$. *Let* $L_i$ *be the score defined above. The prediction set:*

$$\mathcal{C}^{\text{CMR}}(x_{n+1}) = \{y \in \mathbb{R} \mid L(y; x_{n+1}) \leq \widehat{q}_{1-\alpha}\}$$

*satisfies* $\mathbb{P}(y_{n+1} \in \mathcal{C}^{\text{CMR}}(x_{n+1})) \geq 1 - \alpha$.

### 3.4 Prediction Interval Construction

For a test input $x_{n+1}$, we generate and sort $m$ samples. The prediction interval is constructed by selecting the shortest interval that contains $\widehat{q}_{1-\alpha}$ samples:

$$\mathcal{C}^{\text{CMR}}(x_{n+1}) = \left[y_{n+1}^{(j^*)}, y_{n+1}^{(j^*+\widehat{q}_{1-\alpha}-1)}\right], \tag{14}$$

where $j^*$ minimizes the interval width.

---

**Algorithm 1** Conditional Flow Matching for Conformal Regression (CMR)

---

**Require:** Dataset $\mathcal{D}$, noise $\sigma$, samples $m$, confidence $\alpha$.
**Ensure:** Prediction interval $\mathcal{C}^{\text{CMR}}(x_{n+1})$.
 1: Split $\mathcal{D}$ into $\mathcal{I}_{\text{train}}, \mathcal{I}_{\text{cal}}, \mathcal{I}_{\text{test}}$.
    ▷ *Train Conditional Flow Matching Model*
 2: Initialize $\theta$ for $v_\theta(t, y, x)$.
 3: **for** each training iteration **do**
 4:     Sample batch $\{(x_i, y_i)\}_{i=1}^B$ from $\mathcal{I}_{\text{train}}$.
 5:     Sample $y_0^i \sim \mathcal{N}(0, 1)$ and $t^i \sim \mathcal{U}(0, 1)$.
 6:     Compute path: $y_t^i = t^i y_i + (1 - t^i)y_0^i + \sigma\epsilon$.
 7:     Compute target vector: $u_t^i = y_i - y_0^i$.
 8:     Loss $\mathcal{L} = \frac{1}{B}\sum_{i=1}^B \|v_\theta(t^i, y_t^i, x_i) - u_t^i\|^2$.
 9:     Update $\theta$ via gradient descent.
10: **end for**

    ▷ *Calibration*
11: Initialize scores $L = []$.
12: **for** $(x_i, y_i)$ in $\mathcal{I}_{\text{cal}}$ **do**
13:     Generate $m$ samples $\{y_i^j\}_{j=1}^m$ via ODE solve.
14:     Sort samples to get $\{y_i^{(1)}, \ldots, y_i^{(m)}\}$.
15:     $L_i \leftarrow \text{FindMinSampleCount}(y_i^{(1...m)}, y_i^{true})$.
16: **end for**
17: $\widehat{q}_{1-\alpha} \leftarrow \text{Quantile}_{\lceil(1-\alpha)(|\mathcal{I}_{\text{cal}}|+1)\rceil/|\mathcal{I}_{\text{cal}}|}(L)$.

    ▷ *Inference*
18: **for** $x_{n+1}$ in $\mathcal{I}_{\text{test}}$ **do**
19:     Generate $m$ samples $\{y_{n+1}^{(1...m)}\}$ via ODE solve.
20:     $\mathcal{C}^{\text{CMR}} \leftarrow \text{FindShortestInterval}(y_{n+1}^{(1...m)}, \text{count} = \widehat{q}_{1-\alpha})$.
21: **end for**

---

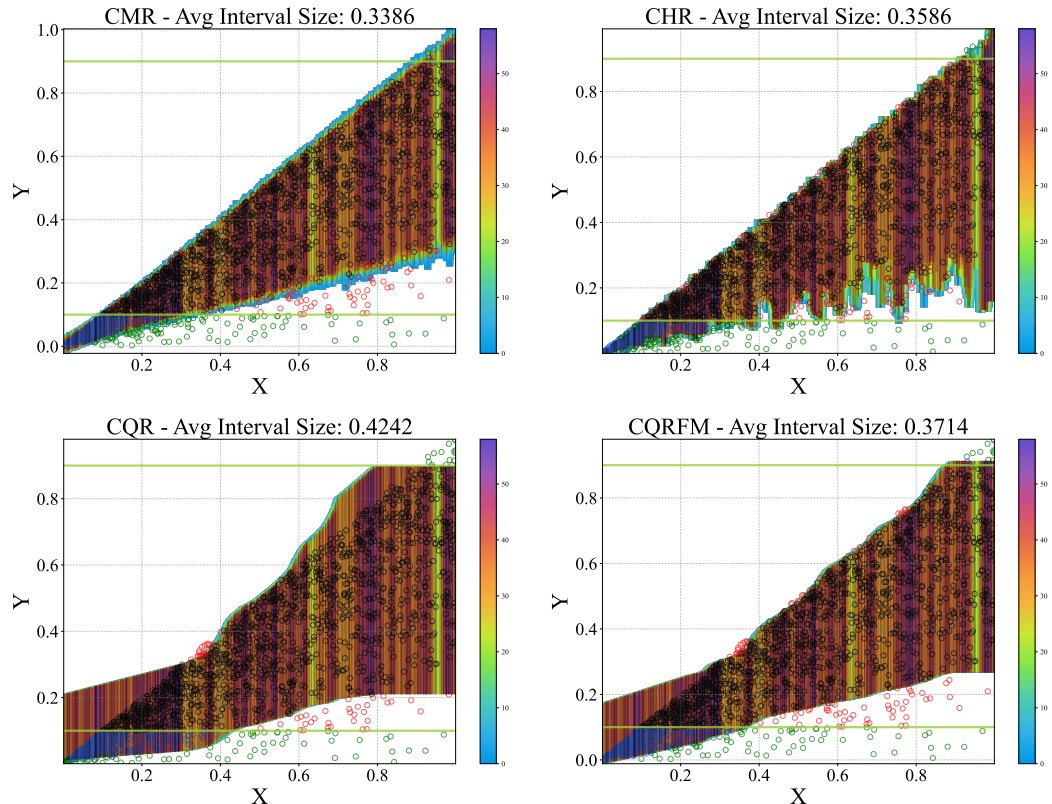

Figure 2: In the scatter plot of features ($x$) vs. true labels ($y$), points are color-coded: **green** (uncovered tail points), **blue** (covered tail points), **red** (uncovered points), and **black** (covered points). All conformal regression methods output prediction intervals visualized as vertical bars (fixed width: 0.01) with heights reflecting interval widths. Taller bars correspond to wider intervals, while overlapping regions (color-coded by density) highlight prediction set sizes: denser overlaps imply larger sets. This visualization contrasts interval characteristics and coverage patterns across methods.

## 4 EXPERIMENTS

In this section, we evaluated our proposed method through simulation studies ($1 - D/2 - D$ scenarios) and experiments on 12 real-world datasets. Performance was measured using Coverage, Size, Worse-Slab Coverage (WSC) (Cauchois et al., 2021) and Tail coverage rate (TCR) (Lin et al., 2021) as key metrics. Comparative analysis included six baseline methods: CHR (Sesia & Romano, 2021), CQR (Romano et al., 2019), CQRR (Sesia & Candès, 2020), CQRM (Kivaranovic et al., 2020), CQRFM (Kivaranovic et al., 2020), and Split Conformal methods (details in Appendix A.4). Experiments used fixed hyperparameters ($m = 1000$, $\sigma = 0.01$, $\alpha = 0.1$) with sensitivity analysis presented in Appendix A.8. All results were validated through 50 randomized experimental partitions to ensure statistical reliability.

### 4.1 EVALUATION METRICS (WSC AND TCR)

In addition to the standard coverage and size metrics, our evaluation framework incorporates Worst-Slab Coverage (WSC) and Tail Coverage Rate (TCR) to provide a more nuanced understanding of prediction interval performance, particularly concerning conditional validity and robustness across data distributions.

### 4.1.1 Worst-Slab Coverage (WSC)

The Worst-Slab Coverage (WSC) metric, introduced by Cauchois et al. (2021), assesses the robustness of coverage guarantees across various subsets of the feature space. It identifies the minimum coverage achieved within "slabs" of the data, thereby highlighting potential regions where a model might underperform in its uncertainty quantification.

A slab $S_{v,a,b}$ is defined as a region in the feature space:

$$S_{v,a,b} := \{x \in \mathbb{R}^d \mid a \leq v^T x \leq b\}$$

where $v \in \mathbb{R}^d$ is a direction vector and $a < b \in \mathbb{R}$ define the bounds of the slab.

The Worst-Slab Coverage $WSC_n(C, v, \delta)$ for a given confidence set $C$, direction $v$, and minimum mass $\delta$ is then formally defined as:

$$WSC_n(C, v, \delta) := \inf_{a<b} \left\{ P_n(Y \in C(X) \mid a \leq v^T X \leq b) \quad \text{s.t.} \quad P_n(a \leq v^T X \leq b) \geq \delta \right\} \quad (15)$$

Here, $P_n$ denotes the empirical distribution on the observed data. This metric quantifies the lowest coverage rate observed in any slab that contains at least a $\delta$ fraction of the data, providing insight into the conditional performance of the prediction intervals.

### 4.1.2 Tail Coverage Rate (TCR)

The Tail Coverage Rate (TCR) is a crucial metric for evaluating how well prediction intervals perform on extreme values of the response variable. As described by Lin et al. (2021), it specifically measures the coverage rate for data points whose true labels fall within the top and bottom 10% of the response distribution.

Let $Y_{\text{test}} = \{Y_1, \ldots, Y_N\}$ be the set of true labels in the test set, and $C(X_i)$ be the prediction interval constructed for the input $X_i$. To calculate the TCR, we first identify the thresholds for the extreme tails of the $Y_{\text{test}}$ distribution. Let $Q_{0.1}$ be the 10th percentile and $Q_{0.9}$ be the 90th percentile of $Y_{\text{test}}$.

The set of tail examples $I_{\text{tail}}$ is then defined as:

$$I_{\text{tail}} = \{i \mid Y_i \leq Q_{0.1} \quad \text{or} \quad Y_i \geq Q_{0.9}\}$$

The Tail Coverage Rate (TCR) is subsequently calculated as the proportion of these tail examples whose true labels are contained within their respective prediction intervals:

$$\text{TCR} = \frac{1}{|I_{\text{tail}}|} \sum_{i \in I_{\text{tail}}} \mathbf{1}_{Y_i \in C(X_i)} \quad (16)$$

This metric is particularly important in applications where accurate uncertainty quantification for extreme outcomes is critical.

### 4.2 Simulation Study

We conducted experiments on both 1D and 2D simulated data. Detailed parameter settings can be found in Appendix A.5 and A.6. We have visualized the method with the smallest size (CHR(RF), CQRFM) and a classic approach (CQR). As noted in the caption of Figure 2, this visualization comprehensively reflects four key metrics - coverage, size, WSC, and TCR - across different conformal regression methods (Additional information: For details about the WSC and TCR indicators, please refer to Appendix 4.1). It is particularly worth emphasizing that the size dimension can be interpreted through both density variations and the spatial extent of coverage areas. The visualization clearly demonstrates that CMR exhibits the narrowest coverage region, followed by CHR, while CQR and CHRFM show substantial wasted coverage in the lower-left quadrant. We have also conducted visual analysis on the 2D data, with detailed results provided in Appendix A.6.

## 4.3 Real Data

We conducted experiments on 12 real-world datasets (dat, h;i;j;c;b;a;d; Achilles et al., 2008; dat, e;f;g). Following the methodology outlined in Sesia & Candès (2020), we rescale the response $Y$ by the mean absolute value. We randomly allocate 20% of the samples for testing, and from the remaining data, we utilize 70% for training the quantile regression model and 30% for calibration.

| Dataset | Metric | CMR(CFM) | CHR(RF) | CHR(NN) | CQR | CQRM | CQRR | CQRFM | Split |
|---|---|---|---|---|---|---|---|---|---|
| synthetic[1] | Coverage | 0.901 (0.007) | 0.897 (0.007) | 0.900 (0.009) | 0.900 (0.010) | 0.901 (0.010) | 0.899 (0.009) | 0.899 (0.011) | 0.900 (0.010) |
| | Size | 0.350 (0.007) | 0.361 (0.008) | 0.369 (0.014) | 0.442 (0.020) | 0.402 (0.020) | 0.434 (0.018) | 0.403 (0.018) | 0.475 (0.020) |
| | TCR | 0.865 (0.020) | 0.850 (0.021) | 0.833 (0.043) | 0.722 (0.072) | 0.804 (0.060) | 0.744 (0.068) | 0.776 (0.080) | 0.764 (0.070) |
| synthetic[2] | Coverage | 0.897 (0.008) | 0.893 (0.026) | 0.899 (0.009) | 0.901 (0.009) | 0.901 (0.009) | 0.900 (0.010) | 0.902 (0.009) | 0.900 (0.009) |
| | Size | 1.023 (0.019) | 1.051 (0.016) | 1.063 (0.018) | 1.157 (0.042) | 1.113 (0.041) | 1.161 (0.028) | 1.119 (0.035) | 1.190 (0.041) |
| | TCR | 0.728 (0.025) | 0.748 (0.026) | 0.713 (0.029) | 0.677 (0.051) | 0.637 (0.057) | 0.665 (0.050) | 0.633 (0.046) | 0.544 (0.053) |
| bike | Coverage | 0.902 (0.009) | 0.902 (0.008) | 0.900 (0.009) | 0.904 (0.010) | 0.903 (0.008) | 0.905 (0.009) | 0.901 (0.008) | 0.900 (0.010) |
| | Size | 0.521 (0.016) | 1.128 (0.041) | 0.757 (0.043) | 1.564 (0.086) | 1.289 (0.089) | 1.534 (0.078) | 1.313 (0.093) | 1.355 (0.094) |
| | WSC | 0.883 (0.035) | 0.879 (0.041) | 0.885 (0.030) | 0.797 (0.065) | 0.831 (0.049) | 0.813 (0.057) | 0.832 (0.053) | 0.821 (0.049) |
| | TCR | 0.927 (0.015) | 0.837 (0.019) | 0.844 (0.021) | 0.621 (0.058) | 0.7000 (0.055) | 0.650 (0.060) | 0.695 (0.048) | 0.709 (0.041) |
| bio | Coverage | 0.899 (0.004) | 0.899 (0.004) | 0.898 (0.005) | 0.899 (0.005) | 0.900 (0.004) | 0.899 (0.005) | 0.899 (0.004) | 0.899 (0.004) |
| | Size | 1.162 (0.019) | 1.456 (0.019) | 1.578 (0.022) | 2.007 (0.024) | 1.984 (0.034) | 2.009 (0.029) | 1.980 (0.041) | 1.982 (0.053) |
| | WSC | 0.879 (0.017) | 0.883 (0.020) | 0.877 (0.015) | 0.805 (0.025) | 0.812 (0.022) | 0.805 (0.025) | 0.814 (0.030) | 0.834 (0.032) |
| | TCR | 0.842 (0.010) | 0.749 (0.010) | 0.724 (0.012) | 0.558 (0.026) | 0.574 (0.028) | 0.556 (0.025) | 0.570 (0.027) | 0.731 (0.033) |
| blog | Coverage | 0.900 (0.004) | 0.901 (0.004) | 0.901 (0.005) | 0.942 (0.008) | 0.908 (0.011) | 0.942 (0.008) | 0.909 (0.018) | 0.909 (0.005) |
| | Size | 1.362 (0.048) | 1.596 (0.125) | 1.771 (0.171) | 3.340 (0.359) | 1.943 (0.360) | 3.303 (0.360) | 2.053 (0.648) | 1.429 (0.105) |
| | WSC | 0.770 (0.018) | 0.862 (0.022) | 0.859 (0.022) | 0.866 (0.029) | 0.751 (0.039) | 0.866 (0.035) | 0.766 (0.050) | 0.676 (0.025) |
| | TCR | 0.699 (0.007) | 0.266 (0.017) | 0.249 (0.015) | 0.251 (0.019) | 0.573 (0.151) | 0.297 (0.147) | 0.652 (0.048) | 0.101 (0.141) |
| community | Coverage | 0.898 (0.023) | 0.899 (0.022) | 0.896 (0.019) | 0.905 (0.023) | 0.900 (0.021) | 0.903 (0.022) | 0.899 (0.023) | 0.899 (0.021) |
| | Size | 1.566 (0.080) | 1.636 (0.116) | 1.574 (0.104) | 1.758 (0.133) | 1.661 (0.116) | 1.764 (0.116) | 1.674 (0.104) | 2.183 (0.224) |
| | WSC | 0.880 (0.080) | 0.892 (0.057) | 0.895 (0.057) | 0.889 (0.068) | 0.889 (0.051) | 0.901 (0.059) | 0.885 (0.052) | 0.775 (0.115) |
| | TCR | 0.813 (0.048) | 0.713 (0.053) | 0.771 (0.041) | 0.698 (0.094) | 0.750 (0.064) | 0.658 (0.102) | 0.771 (0.066) | 0.670 (0.062) |
| concrete | Coverage | 0.898 (0.028) | 0.903 (0.031) | 0.907 (0.024) | 0.900 (0.026) | 0.902 (0.024) | 0.904 (0.025) | 0.903 (0.022) | 0.905 (0.026) |
| | Size | 0.483 (0.043) | 0.942 (0.065) | 0.493 (0.036) | 0.700 (0.055) | 0.628 (0.050) | 0.699 (0.053) | 0.629 (0.056) | 0.607 (0.051) |
| | WSC | 0.848 (0.157) | 0.879 (0.107) | 0.888 (0.088) | 0.883 (0.094) | 0.882 (0.158) | 0.884 (0.155) | 0.918 (0.091) | 0.859 (0.119) |
| | TCR | 0.890 (0.051) | 0.725 (0.098) | 0.880 (0.058) | 0.743 (0.091) | 0.799 (0.075) | 0.768 (0.081) | 0.811 (0.081) | 0.802 (0.080) |
| facebook1 | Coverage | 0.900 (0.004) | 0.901 (0.004) | 0.901 (0.005) | 0.943 (0.013) | 0.913 (0.022) | 0.944 (0.013) | 0.918 (0.027) | 0.902 (0.005) |
| | Size | 1.176 (0.045) | 1.559 (0.103) | 1.389 (0.081) | 2.693 (0.466) | 2.148 (0.979) | 2.696 (0.345) | 2.367 (1.565) | 2.156 (0.188) |
| | WSC | 0.746 (0.032) | 0.838 (0.025) | 0.838 (0.030) | 0.878 (0.033) | 0.819 (0.058) | 0.879 (0.035) | 0.830 (0.063) | 0.519 (0.032) |
| | TCR | 0.789 (0.012) | 0.356 (0.013) | 0.339 (0.012) | 0.418 (0.024) | 0.835 (0.119) | 0.458 (0.013) | 0.880 (0.038) | 0.540 (0.014) |
| facebook2 | Coverage | 0.899 (0.003) | 0.899 (0.003) | 0.899 (0.003) | 0.949 (0.008) | 0.910 (0.020) | 0.935 (0.076) | 0.909 (0.016) | 0.902 (0.003) |
| | Size | 1.201 (0.050) | 1.547 (0.059) | 1.413 (0.068) | 2.778 (0.424) | 1.910 (0.414) | 2.747 (0.597) | 1.847 (0.211) | 2.140 (0.177) |
| | WSC | 0.756 (0.022) | 0.826 (0.016) | 0.831 (0.015) | 0.878 (0.029) | 0.803 (0.050) | 0.854 (0.110) | 0.805 (0.035) | 0.518 (0.023) |
| | TCR | 0.793 (0.014) | 0.345 (0.009) | 0.344 (0.008) | 0.421 (0.020) | 0.825 (0.125) | 0.456 (0.129) | 0.871 (0.021) | 0.532 (0.013) |
| homes | Coverage | 0.901 (0.006) | 0.900 (0.006) | 0.895 (0.005) | 0.900 (0.006) | 0.901 (0.007) | 0.899 (0.005) | 0.899 (0.007) | 0.897 (0.006) |
| | Size | 0.521 (0.014) | 0.684 (0.017) | 0.538 (0.010) | 0.847 (0.055) | 0.728 (0.034) | 0.826 (0.047) | 0.734 (0.037) | 0.829 (0.077) |
| | WSC | 0.884 (0.025) | 0.834 (0.038) | 0.883 (0.018) | 0.746 (0.067) | 0.805 (0.049) | 0.758 (0.062) | 0.799 (0.053) | 0.597 (0.051) |
| | TCR | 0.838 (0.021) | 0.797 (0.016) | 0.851 (0.014) | 0.689 (0.048) | 0.729 (0.041) | 0.698 (0.045) | 0.718 (0.045) | 0.640 (0.036) |
| meps19 | Coverage | 0.900 (0.008) | 0.902 (0.007) | 0.900 (0.007) | 0.928 (0.009) | 0.916 (0.010) | 0.927 (0.010) | 0.918 (0.009) | 0.902 (0.008) |
| | Size | 2.288 (0.146) | 2.333 (0.163) | 2.547 (0.151) | 2.898 (0.177) | 2.599 (0.168) | 2.908 (0.190) | 2.622 (0.146) | 3.061 (0.274) |
| | WSC | 0.855 (0.042) | 0.890 (0.025) | 0.897 (0.021) | 0.863 (0.034) | 0.820 (0.045) | 0.860 (0.043) | 0.832 (0.046) | 0.592 (0.059) |
| | TCR | 0.729 (0.024) | 0.252 (0.020) | 0.296 (0.018) | 0.237 (0.028) | 0.280 (0.147) | 0.278 (0.130) | 0.708 (0.022) | 0.575 (0.031) |
| meps20 | Coverage | 0.901 (0.007) | 0.902 (0.006) | 0.902 (0.006) | 0.927 (0.008) | 0.920 (0.009) | 0.928 (0.008) | 0.918 (0.009) | 0.902 (0.006) |
| | Size | 2.146 (0.097) | 2.357 (0.148) | 2.515 (0.118) | 2.879 (0.136) | 2.745 (0.164) | 2.942 (0.147) | 2.718 (0.127) | 3.052 (0.289) |
| | WSC | 0.847 (0.029) | 0.893 (0.025) | 0.896 (0.017) | 0.856 (0.038) | 0.834 (0.041) | 0.862 (0.036) | 0.821 (0.044) | 0.606 (0.047) |
| | TCR | 0.722 (0.018) | 0.237 (0.017) | 0.282 (0.015) | 0.227 (0.027) | 0.277 (0.134) | 0.238 (0.072) | 0.696 (0.021) | 0.573 (0.035) |
| meps21 | Coverage | 0.902 (0.008) | 0.902 (0.007) | 0.902 (0.008) | 0.930 (0.007) | 0.922 (0.009) | 0.929 (0.007) | 0.920 (0.008) | 0.903 (0.007) |
| | Size | 2.116 (0.112) | 2.474 (0.156) | 2.667 (0.179) | 2.927 (0.149) | 2.714 (0.134) | 2.934 (0.196) | 2.689 (0.164) | 2.948 (0.245) |
| | WSC | 0.854 (0.044) | 0.896 (0.022) | 0.895 (0.021) | 0.854 (0.029) | 0.826 (0.042) | 0.854 (0.037) | 0.833 (0.035) | 0.618 (0.065) |
| | TCR | 0.718 (0.019) | 0.234 (0.016) | 0.289 (0.018) | 0.211 (0.019) | 0.239 (0.113) | 0.219 (0.071) | 0.685 (0.021) | 0.567 (0.031) |
| star | Coverage | 0.900 (0.019) | 0.900 (0.019) | 0.902 (0.019) | 0.905 (0.020) | 0.904 (0.018) | 0.901 (0.017) | 0.900 (0.023) | 0.901 (0.022) |
| | Size | 0.179 (0.005) | 0.179 (0.006) | 0.207 (0.007) | 0.181 (0.006) | 0.181 (0.006) | 0.181 (0.006) | 0.178 (0.007) | 0.177 (0.006) |
| | WSC | 0.914 (0.052) | 0.892 (0.050) | 0.901 (0.057) | 0.894 (0.056) | 0.893 (0.056) | 0.900 (0.058) | 0.900 (0.054) | 0.895 (0.059) |
| | TCR | 0.638 (0.070) | 0.607 (0.071) | 0.724 (0.059) | 0.578 (0.079) | 0.584 (0.077) | 0.571 (0.070) | 0.573 (0.090) | 0.603 (0.090) |

Table 1: The coverage, size, WSC and TCR results for various methods are presented in the table. For various indicators, the top-ranked method is highlighted in bold bluee text, while the second-ranked method is displayed in bold green.

As shown in Table 1, CMR achieves the smallest size across all comparative methods in both simulation datasets and 11 real-world datasets. Furthermore, its TCR consistently ranks first or second across most datasets. Notably, CMR demonstrates substantial superiority in tail coverage performance on the "meps19" series and Blog datasets compared to baseline methods. However, on the "star" dataset ($n = 2161$) with limited sample size, CMR exhibits a slightly larger interval size (0.02 higher than the optimal benchmark), suggesting potential challenges in fully capturing data distribution patterns under extreme sample scarcity. These results underscore the critical role of base model selection in conformal prediction performance, with CMR exhibiting remarkable competitive advantages. Boxplots of key evaluation metrics are provided in Appendix A.7 for further visualization.

## 4.4 Ablation Experiment

| Dataset | Metric | CMR(CFM) | CMR(RF) | CMR(NN) | CHR(CFM) | CHR(RF) | CHR(NN) | CQR(CFM) | CQR |
|---|---|---|---|---|---|---|---|---|---|
| synthetic[1] | Coverage | 0.901 (0.007) | 0.898 (0.007) | 0.902 (0.013) | 0.905 (0.010) | 0.897 (0.007) | 0.900 (0.009) | 0.904 (0.009) | 0.900 (0.010) |
| | Size | **0.350 (0.007)** | 0.356 (0.004) | 0.355 (0.009) | **0.356 (0.027)** | 0.361 (0.008) | 0.369 (0.014) | **0.383 (0.012)** | 0.442 (0.020) |
| | TCR | **0.865 (0.020)** | 0.832 (0.015) | 0.852 (0.026) | **0.880 (0.031)** | 0.850 (0.021) | 0.833 (0.043) | 0.819 (0.035) | 0.722 (0.072) |
| synthetic[2] | Coverage | 0.897 (0.008) | 0.896 (0.005) | 0.899 (0.012) | 0.904 (0.008) | 0.893 (0.026) | 0.899 (0.009) | 0.898 (0.008) | 0.901 (0.009) |
| | Size | **1.023 (0.019)** | 1.031 (0.014) | 1.024 (0.029) | 1.073 (0.026) | **1.051 (0.016)** | 1.063 (0.018) | **1.134 (0.047)** | 1.157 (0.042) |
| | TCR | **0.728 (0.025)** | 0.734 (0.039) | 0.691 (0.043) | 0.711 (0.035) | **0.748 (0.026)** | 0.713 (0.029) | 0.733 (0.081) | 0.677 (0.051) |
| bike | Coverage | 0.902 (0.009) | 0.899 (0.007) | 0.902 (0.005) | 0.901 (0.010) | 0.902 (0.008) | 0.900 (0.009) | 0.900 (0.012) | 0.904 (0.010) |
| | Size | **0.521 (0.016)** | 1.024 (0.040) | 0.745 (0.041) | 0.618 (0.056) | 1.128 (0.041) | 0.757 (0.022) | 0.595 (0.028) | 1.564 (0.086) |
| | WSC | 0.883 (0.035) | 0.867 (0.027) | 0.875 (0.049) | **0.892 (0.011)** | 0.879 (0.041) | 0.885 (0.030) | 0.891 (0.013) | 0.797 (0.065) |
| | TCR | **0.927 (0.015)** | 0.815 (0.030) | 0.891 (0.019) | 0.894 (0.003) | 0.837 (0.019) | 0.844 (0.021) | 0.898 (0.003) | 0.621 (0.058) |
| bio | Coverage | 0.899 (0.004) | 0.891 (0.005) | 0.898 (0.005) | 0.899 (0.003) | 0.899 (0.004) | 0.898 (0.005) | 0.901 (0.004) | 0.899 (0.005) |
| | Size | **1.162 (0.019)** | 1.368 (0.020) | 1.578 (0.016) | 1.469 (0.011) | **1.456 (0.019)** | 1.578 (0.022) | **1.606 (0.059)** | 2.007 (0.024) |
| | WSC | 0.879 (0.017) | 0.878 (0.022) | 0.881 (0.020) | 0.894 (0.004) | 0.883 (0.020) | 0.877 (0.015) | **0.895 (0.005)** | 0.805 (0.025) |
| | TCR | **0.842 (0.010)** | 0.752 (0.009) | 0.758 (0.008) | 0.788 (0.018) | 0.749 (0.010) | 0.724 (0.012) | **0.721 (0.029)** | 0.558 (0.026) |
| blog | Coverage | 0.900 (0.004) | 0.903 (0.004) | 0.901 (0.002) | 0.900 (0.003) | 0.901 (0.004) | 0.901 (0.005) | 0.900 (0.002) | 0.942 (0.008) |
| | Size | **1.362 (0.048)** | 1.712 (0.088) | 1.997 (0.182) | **1.571 (0.189)** | 1.596 (0.125) | 1.771 (0.171) | 2.646 (0.494) | 3.340 (0.359) |
| | WSC | 0.770 (0.018) | **0.886 (0.017)** | 0.878 (0.028) | **0.904 (0.004)** | 0.862 (0.022) | 0.859 (0.022) | **0.905 (0.003)** | 0.751 (0.039) |
| | TCR | 0.699 (0.007) | 0.296 (0.010) | **0.770 (0.015)** | **0.922 (0.006)** | 0.266 (0.017) | 0.249 (0.015) | **0.903 (0.012)** | 0.251 (0.019) |
| community | Coverage | 0.898 (0.023) | 0.909 (0.019) | 0.897 (0.020) | 0.904 (0.025) | 0.899 (0.022) | 0.896 (0.019) | 0.901 (0.024) | 0.905 (0.023) |
| | Size | 1.566 (0.080) | **1.545 (0.093)** | 1.565 (0.097) | **1.463 (0.103)** | 1.636 (0.116) | 1.574 (0.104) | 1.619 (0.091) | 1.758 (0.133) |
| | WSC | 0.880 (0.080) | 0.887 (0.048) | **0.896 (0.038)** | 0.895 (0.021) | 0.892 (0.057) | 0.895 (0.057) | **0.897 (0.019)** | 0.889 (0.068) |
| | TCR | 0.813 (0.048) | 0.672 (0.061) | **0.838 (0.039)** | 0.824 (0.042) | 0.713 (0.053) | 0.771 (0.041) | **0.811 (0.039)** | 0.698 (0.094) |
| concrete | Coverage | 0.898 (0.028) | 0.909 (0.016) | 0.905 (0.015) | 0.903 (0.028) | 0.903 (0.031) | 0.907 (0.024) | 0.905 (0.032) | 0.900 (0.026) |
| | Size | 0.483 (0.043) | **0.900 (0.060)** | **0.470 (0.054)** | 0.539 (0.019) | 0.942 (0.065) | **0.493 (0.036)** | 0.538 (0.038) | 0.700 (0.055) |
| | WSC | 0.848 (0.157) | **0.875 (0.092)** | 0.870 (0.102) | **0.890 (0.024)** | 0.879 (0.107) | 0.888 (0.088) | 0.885 (0.035) | 0.883 (0.094) |
| | TCR | 0.890 (0.051) | 0.749 (0.065) | **0.929 (0.038)** | 0.898 (0.026) | 0.725 (0.098) | 0.880 (0.058) | 0.883 (0.048) | 0.743 (0.091) |
| facebook1 | Coverage | 0.900 (0.004) | 0.903 (0.005) | 0.899 (0.006) | 0.902 (0.005) | 0.901 (0.004) | 0.901 (0.005) | 0.900 (0.004) | 0.943 (0.013) |
| | Size | **1.176 (0.045)** | 1.600 (0.055) | 1.572 (0.093) | **1.385 (0.013)** | 1.559 (0.103) | 1.389 (0.081) | 2.093 (0.157) | 2.693 (0.466) |
| | WSC | 0.746 (0.032) | 0.879 (0.022) | **0.884 (0.014)** | **0.901 (0.004)** | 0.838 (0.025) | 0.838 (0.030) | **0.897 (0.005)** | 0.878 (0.033) |
| | TCR | 0.789 (0.012) | 0.369 (0.011) | **0.865 (0.010)** | **0.943 (0.005)** | 0.356 (0.013) | 0.339 (0.012) | **0.934 (0.012)** | 0.418 (0.024) |
| facebook2 | Coverage | 0.899 (0.003) | 0.900 (0.002) | 0.900 (0.003) | 0.899 (0.002) | 0.899 (0.003) | 0.899 (0.003) | 0.898 (0.003) | 0.949 (0.008) |
| | Size | **1.201 (0.050)** | 1.521 (0.056) | 1.562 (0.051) | **1.412 (0.052)** | 1.547 (0.059) | 1.413 (0.068) | 2.146 (0.109) | 2.778 (0.424) |
| | WSC | 0.756 (0.022) | 0.871 (0.011) | **0.889 (0.017)** | **0.897 (0.003)** | 0.826 (0.016) | 0.831 (0.015) | **0.893 (0.003)** | 0.878 (0.029) |
| | TCR | 0.793 (0.014) | 0.361 (0.007) | **0.876 (0.007)** | **0.945 (0.004)** | 0.345 (0.009) | 0.344 (0.008) | **0.932 (0.011)** | 0.421 (0.020) |
| homes | Coverage | 0.901 (0.006) | 0.903 (0.005) | 0.899 (0.009) | 0.900 (0.007) | 0.900 (0.006) | 0.895 (0.005) | 0.899 (0.005) | 0.900 (0.006) |
| | Size | **0.521 (0.014)** | 0.682 (0.016) | 0.527 (0.009) | 0.691 (0.146) | 0.684 (0.017) | 0.538 (0.010) | 0.759 (0.270) | 0.847 (0.055) |
| | WSC | 0.884 (0.025) | 0.830 (0.030) | **0.884 (0.021)** | **0.896 (0.008)** | 0.834 (0.038) | 0.883 (0.018) | **0.895 (0.006)** | 0.746 (0.067) |
| | TCR | **0.838 (0.021)** | 0.797 (0.012) | 0.837 (0.013) | 0.780 (0.010) | 0.797 (0.016) | **0.851 (0.014)** | 0.779 (0.012) | 0.689 (0.048) |
| meps19 | Coverage | 0.900 (0.008) | 0.902 (0.007) | 0.902 (0.004) | 0.895 (0.005) | 0.902 (0.007) | 0.900 (0.007) | 0.899 (0.004) | 0.928 (0.009) |
| | Size | **2.288 (0.146)** | 2.380 (0.160) | 2.602 (0.160) | **2.280 (0.311)** | 2.333 (0.163) | 2.547 (0.151) | 2.853 (0.246) | 2.898 (0.177) |
| | WSC | 0.855 (0.042) | **0.897 (0.026)** | 0.865 (0.056) | 0.891 (0.005) | 0.890 (0.025) | 0.897 (0.021) | **0.895 (0.004)** | 0.863 (0.034) |
| | TCR | 0.729 (0.024) | 0.258 (0.018) | **0.815 (0.044)** | **0.855 (0.013)** | 0.252 (0.020) | 0.296 (0.018) | 0.838 (0.018) | 0.237 (0.028) |
| meps20 | Coverage | 0.901 (0.007) | 0.901 (0.007) | 0.901 (0.005) | 0.905 (0.005) | 0.902 (0.006) | 0.902 (0.006) | 0.902 (0.005) | 0.927 (0.008) |
| | Size | **2.146 (0.097)** | 2.321 (0.152) | 2.586 (0.105) | **2.288 (0.181)** | 2.357 (0.148) | 2.515 (0.118) | 3.046 (0.224) | **2.879 (0.136)** |
| | WSC | 0.847 (0.029) | 0.896 (0.022) | **0.901 (0.016)** | **0.900 (0.004)** | 0.893 (0.025) | 0.896 (0.017) | **0.896 (0.005)** | 0.856 (0.038) |
| | TCR | 0.722 (0.018) | 0.226 (0.015) | **0.767 (0.015)** | 0.833 (0.022) | 0.237 (0.017) | 0.282 (0.015) | **0.833 (0.014)** | 0.227 (0.027) |
| meps21 | Coverage | 0.902 (0.008) | 0.906 (0.005) | 0.898 (0.006) | 0.903 (0.008) | 0.902 (0.007) | 0.902 (0.008) | 0.902 (0.003) | 0.930 (0.007) |
| | Size | **2.116 (0.112)** | 2.340 (0.143) | 2.580 (0.107) | **2.116 (0.113)** | 2.474 (0.156) | 2.667 (0.179) | 2.763 (0.240) | 2.927 (0.149) |
| | WSC | 0.854 (0.044) | **0.895 (0.026)** | 0.887 (0.020) | **0.899 (0.005)** | 0.896 (0.022) | 0.895 (0.021) | **0.897 (0.003)** | 0.854 (0.029) |
| | TCR | 0.718 (0.019) | 0.223 (0.014) | **0.769 (0.017)** | **0.843 (0.017)** | 0.234 (0.016) | 0.289 (0.018) | 0.842 (0.014) | 0.211 (0.019) |
| star | Coverage | 0.900 (0.019) | 0.899 (0.017) | 0.909 (0.020) | 0.904 (0.018) | 0.900 (0.019) | 0.902 (0.019) | 0.905 (0.023) | 0.905 (0.020) |
| | Size | **0.179 (0.005)** | 0.179 (0.006) | 0.201 (0.008) | 0.201 (0.011) | **0.179 (0.006)** | 0.207 (0.007) | 0.211 (0.015) | **0.181 (0.006)** |
| | WSC | **0.914 (0.052)** | 0.913 (0.052) | 0.892 (0.045) | 0.888 (0.021) | 0.892 (0.050) | **0.901 (0.057)** | 0.889 (0.025) | **0.894 (0.056)** |
| | TCR | 0.638 (0.070) | 0.607 (0.064) | **0.745 (0.061)** | **0.755 (0.049)** | 0.607 (0.071) | 0.724 (0.059) | **0.731 (0.055)** | 0.578 (0.079) |

Table 2: The coverage, size, WSC and TCR results for various methods are presented in the table. An orange background indicates the best metric within the dataset. A blue background denotes the better metric between CMR and CHR when using the same underlying predictive model. Additionally, bold text represents the better metric among different underlying models under the same non-conformity score condition (CMR, CHR, or CQR).

In this section, we will decouple CMR to demonstrate how the non-conformity scores of CFM and CMR jointly enhance the efficiency of prediction sets. Specifically, we conducted comparative experiments where we tested neural networks and random forests as base predictive models respectively, and also tested CFM as the base predictive model, comparing their performances across three non-conformity score functions: CMR, CHR, and CQR. Each dataset was randomly split 10 times.

As shown in Table 2, we use an orange background to indicate the best metric within the dataset, and a blue background to denote the better metric between CMR and CHR under the same base predictive model. Bold text is used to highlight the better metric among different base models under the same non-conformity score condition (CMR, CHR, CQR). It can be observed that the optimal metrics are either achieved by using CMR as the non-conformity score or by using CFM as the base predictive model. Secondly, the prediction set sizes generated by CMR (with CFM as the base model) are optimal in 11 out of 14 datasets. Using CFM as the base model ensures stable conditional coverage and TCR. Compared to CHR, CMR generally outperforms CHR across three different base predictive models. Furthermore, we have also analyzed the resource consumption of CFM under different calibration methods, with details provided in Appendix A.9.

## 5 CONCLUSION

We have introduced Conditional Flow Matching for Conformal Regression (CMR), a novel framework that effectively combines conditional flow matching with conformal prediction to generate statistically valid prediction intervals for regression tasks. Our approach employs a simulation-free training method to learn the conditional distribution of the target variable and then applies a calibration procedure to construct prediction intervals with reduced widths. Our method offers a brand-new perspective for one-dimensional regression tasks, generating continuous prediction intervals that are not only confidence-guaranteed but also the most efficient.

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

# A  APPENDIX

LARGE LANGUAGE MODEL (LLM) USAGE

During the preparation of this manuscript, a large language model (LLM), specifically [Gemini-2.5-Flash], was employed as a general-purpose assist tool. The LLM's contributions were primarily in the following areas:

- **Code Optimization for Experimental Visualization**: The LLM assisted in optimizing and refining Python code snippets used for experimental visualization and data processing routines. This collaboration led to more efficient and readable implementations, particularly for generating the figures and tables presented in the paper.
- **Writing Assistance and Refinement**: The LLM was utilized for drafting and refining certain sections of the paper, including improving clarity, grammar, and stylistic coherence. This involved generating initial textual descriptions and polishing existing content to enhance its overall quality and readability.

The authors maintained full responsibility for reviewing, editing, and validating all content generated or optimized with the assistance of the LLM, ensuring its accuracy, originality, and adherence to scientific standards. The LLM was not involved in the core research ideation, experimental design, data collection, or primary analysis leading to the scientific conclusions. The scientific content, conclusions, and any potential errors remain solely the responsibility of the authors.

## A.1  THEORETICAL ANALYSIS

## A.2  PROOF OF THEOREM 1

**Proof:** Let $n$ be the size of the calibration set $\mathcal{I}_{\text{cal}}$. We consider $n$ calibration points $(x_1, y_1), \ldots, (x_n, y_n)$ and one test point $(x_{n+1}, y_{n+1})$. All these $n+1$ data points are assumed to be drawn independently and identically distributed (i.i.d.) from the same underlying distribution $P$.

For each data point $(x_i, y_i)$, we compute its nonconformity score $L_i$ using the procedure described in Section 3.3. Specifically, $L_i$ is defined as the minimum number of samples $s$ such that the shortest interval constructed from $s$ generated samples contains the true outcome $y_i^{\text{true}}$. Let $L_{n+1}$ denote the nonconformity score for the test point $(x_{n+1}, y_{n+1})$, i.e., $L_{n+1} = L(y_{n+1}; x_{n+1})$.

Since the data points $(x_1, y_1), \ldots, (x_n, y_n), (x_{n+1}, y_{n+1})$ are i.i.d., and the nonconformity score function $L(\cdot; \cdot)$ is applied identically to each point (after the model is trained on a separate training set, if applicable), the resulting nonconformity scores $L_1, \ldots, L_n, L_{n+1}$ are also i.i.d., and therefore **exchangeable**.

Let $k_{cal} = \lceil (n+1)(1-\alpha) \rceil$. The threshold $\widehat{q}_{1-\alpha}$ is defined as the $k_{cal}$-th order statistic of the calibration scores $\{L_1, \ldots, L_n\}$. We denote this as $\widehat{q} = L_{(k_{cal})}$. Note that in our method, $\widehat{q}$ represents a discrete count of samples.

The prediction set for $x_{n+1}$ is $\mathcal{C}^{\text{CMR}}(x_{n+1}) = \{y' \mid L(y'; x_{n+1}) \leq \widehat{q}\}$. We want to show that $\mathbb{P}\left(y_{n+1} \in \mathcal{C}^{\text{CMR}}(x_{n+1})\right) \geq 1 - \alpha$. This is equivalent to showing $\mathbb{P}(L_{n+1} \leq \widehat{q}) \geq 1 - \alpha$.

The event $y_{n+1} \notin \mathcal{C}^{\text{CMR}}(x_{n+1})$ occurs if and only if its nonconformity score $L_{n+1}$ is strictly greater than the threshold $\widehat{q}$:

$$y_{n+1} \notin \mathcal{C}^{\text{CMR}}(x_{n+1}) \iff L_{n+1} > \widehat{q}$$

The event $L_{n+1} > \widehat{q} = L_{(k_{cal})}$ means that $L_{n+1}$ is strictly larger than at least $n - k_{cal} + 1$ of the scores from the calibration set $\{L_1, \ldots, L_n\}$. This implies that the rank of $L_{n+1}$ within the full set of $n + 1$ scores $\{L_1, \ldots, L_n, L_{n+1}\}$ must be at least $k_{cal} + 1$.

Therefore, the probability of non-coverage is bounded by the probability of this rank condition:

$$\mathbb{P}(y_{n+1} \notin \mathcal{C}^{\mathrm{CMR}}(x_{n+1})) = \mathbb{P}(L_{n+1} > \widehat{q}) \tag{17}$$

$$\leq \mathbb{P}(\mathrm{rank}(L_{n+1}) \geq k_{cal} + 1) \tag{18}$$

Since the scores $L_1, \ldots, L_n, L_{n+1}$ are exchangeable, any of the $n+1$ scores is equally likely to take any rank position. Thus, the probability that $L_{n+1}$ has a rank of $r$ (from 1 to $n+1$) is $\frac{1}{n+1}$.

$$\mathbb{P}(\mathrm{rank}(L_{n+1}) \geq k_{cal} + 1) = \sum_{r=k_{cal}+1}^{n+1} \mathbb{P}(\mathrm{rank}(L_{n+1}) = r) = \frac{(n+1) - k_{cal}}{n+1}$$

So, we have:

$$\mathbb{P}(y_{n+1} \notin \mathcal{C}^{\mathrm{CMR}}(x_{n+1})) \leq \frac{(n+1) - k_{cal}}{n+1}$$

Now, we substitute the value of $k_{cal} = \lceil (n+1)(1-\alpha) \rceil$. By the definition of the ceiling function, for any real number $x$, we have $\lceil x \rceil \geq x$. Therefore,

$$k_{cal} \geq (n+1)(1-\alpha)$$

Substituting this into our probability bound:

$$\mathbb{P}(y_{n+1} \notin \mathcal{C}^{\mathrm{CMR}}(x_{n+1})) \leq \frac{n+1 - \lceil (n+1)(1-\alpha) \rceil}{n+1} \tag{19}$$

$$\leq \frac{n+1 - (n+1)(1-\alpha)}{n+1} \tag{20}$$

$$= \frac{(n+1)\alpha}{n+1} = \alpha \tag{21}$$

Finally, the probability of coverage is:

$$\mathbb{P}(y_{n+1} \in \mathcal{C}^{\mathrm{CMR}}(x_{n+1})) = 1 - \mathbb{P}(y_{n+1} \notin \mathcal{C}^{\mathrm{CMR}}(x_{n+1})) \geq 1 - \alpha$$

This proof holds unconditionally over the joint distribution of all $n+1$ samples.

### A.3 Calibrate and test the algorithm

In this section, we provide a detailed description of our algorithm implementation during the calibration and testing phases. By combining the sliding window approach with the binary segmentation algorithm, we have reduced the time complexity of the algorithm in the calibration phase from $O(nm)$ to $O(n \log m)$. Additionally, the time complexity of the testing phase is $O(n)$.

#### A.3.1 Find Minimum Sample Count (Score Calculation)

#### A.3.2 Find Shortest Interval (Inference)

### A.4 Other methods (Conformal Prediction for Regression)

In this section, we provide a detailed exposition of the methodological foundations and operational frameworks underlying the CHR Sesia & Romano (2021), CQR Romano et al. (2019), CQRR Sesia & Candès (2020), CQRM Kivaranovic et al. (2020), and CQRFM Kivaranovic et al. (2020) methodologies.

**Algorithm 2** FINDMINSAMPLECOUNT - Find minimum sample number of interval containing a target value

**Require:** Sorted samples $Y = \{y^{(1)}, y^{(2)}, \ldots, y^{(m)}\}$, target value $y^{true}$.
**Ensure:** Minimum samples $L_{min}$.
 1: Initialize search range: $low \leftarrow 2$, $high \leftarrow m$.
 2: $result\_samples \leftarrow m$ {Default to full range}
 3: **while** $low \leq high$ **do**
 4:     $mid \leftarrow \lfloor (low + high)/2 \rfloor$. {Current sample count $s$}
 5:     $min\_width \leftarrow \infty$.
 6:     $start^* \leftarrow 0, end^* \leftarrow 0$
 7:     **for** $start \in \{0, 1, \ldots, m - mid\}$ **do**
 8:         $end \leftarrow start + mid - 1$.
 9:         $left \leftarrow y^{(start+1)}$, $right \leftarrow y^{(end+1)}$. {Adjust for 1-based index}
10:         $width \leftarrow right - left$.
11:         **if** $width < min\_width$ **then**
12:             $min\_width \leftarrow width$.
13:             $start^* \leftarrow start + 1, end^* \leftarrow end + 1$. {Store indices of shortest interval}
14:         **end if**
15:     **end for**
        {Check if the shortest interval covers the true value}
16:     **if** $y^{(start^*)} \leq y^{true} \leq y^{(end^*)}$ **then**
17:         $result\_samples \leftarrow mid$.
18:         $high \leftarrow mid - 1$.                                                    *Try smaller sample count*
19:     **else**
20:         $low \leftarrow mid + 1$.                                                    *Need larger sample count*
21:     **end if**
22: **end while**
23: Return $result\_samples$

**Algorithm 3** FINDSHORTESTINTERVAL - Find shortest interval with fixed sample count

**Require:** Sorted samples $Y = \{y^{(1)}, y^{(2)}, \ldots, y^{(m)}\}$, sample count threshold $\widehat{q}_{1-\alpha}$ (integer)
**Ensure:** Shortest interval $[y_{low}, y_{up}]$ containing $\widehat{q}_{1-\alpha}$ samples
 1: $min\_width \leftarrow \infty$.
 2: $best\_interval \leftarrow [y^{(1)}, y^{(m)}]$.
 3: $s \leftarrow \widehat{q}_{1-\alpha}$.
 4: **for** $i \in \{1, \ldots, m - s + 1\}$ **do**
 5:     $j \leftarrow i + s - 1$.
 6:     $width \leftarrow y^{(j)} - y^{(i)}$.
 7:     **if** $width < min\_width$ **then**
 8:         $min\_width \leftarrow width$.
 9:         $best\_interval \leftarrow [y^{(i)}, y^{(j)}]$.
10:     **end if**
11: **end for**
12: Return $best\_interval$.

### A.4.1 Conformal Quantile Regression (1).

*Conformal Quantile Regression* constructs intervals based on quantile regression:

$$\mathcal{C}^{\text{CQR}}(x_{n+1}) = \left[\widehat{q}_{\frac{\alpha}{2}}(x_{n+1}) - t_{1-\alpha}^{\text{CQR}}, \widehat{q}_{1-\frac{\alpha}{2}}(x_{n+1}) + t_{1-\alpha}^{\text{CQR}}\right], \tag{22}$$

where $\widehat{q}_{\alpha/2}$ and $\widehat{q}_{1-\alpha/2}$ are conditional quantile estimates, and $t_{1-\alpha}^{\text{CQR}}$ is the $(1-\alpha)(1+1/|\mathcal{I}_{\text{cal}}|)$-th empirical quantile of $\{S_i^{\text{CQR}}\}_{i \in \mathcal{I}_{\text{cal}}} \cup \{\infty\}$, with:

$$S_i^{\text{CQR}} = \max\left(\widehat{q}_{\frac{\alpha}{2}}(x_i) - y_i, y_i - \widehat{q}_{1-\frac{\alpha}{2}}(x_i)\right). \tag{23}$$

### A.4.2 Conformal Quantile Regression (2).

Similar to Sesia & Candès (2020), we name the methods proposed by Kivaranovic et al. (2020) as CQR-m, respectively, which differ from the CQR proposed by Romano et al. (2019). First, let's consider CQR-m. The model defined based on quantile regression is as follows:

$$\mathcal{C}^{\text{CQR-m}}(x_{n+1}) = \left[\widehat{q}_{\frac{\alpha}{2}}(x_{n+1}) - \widehat{\Delta}_{\alpha,\text{lo}}^{\text{CQR-m}}, \widehat{q}_{1-\frac{\alpha}{2}}(x_{n+1}) + \widehat{\Delta}_{\alpha,\text{up}}^{\text{CQR-m}}\right],$$
$$\widehat{\Delta}_{\alpha,\text{lo}}^{\text{CQR-m}} = t_{1-\alpha}^{\text{CQR-m}}\left[\widehat{q}_{\frac{1}{2}}(x_i) - \widehat{q}_{\frac{\alpha}{2}}(x_i)\right], \tag{24}$$
$$\widehat{\Delta}_{\alpha,\text{up}}^{\text{CQR-m}} = t_{1-\alpha}^{\text{CQR-m}}\left[\widehat{q}_{1-\frac{\alpha}{2}}(x_i) - \widehat{q}_{\frac{1}{2}}(x_i)\right],$$

where $\widehat{q}_{1/2}$ indicates an estimated median regression function obtained with the same black-box algorithm as $\widehat{q}_{\alpha/2}$ and $\widehat{q}_{1-\alpha/2}$, and $t_{1-\alpha}^{\text{CQR-m}}$ is the same $(1-\alpha)(1+1/|\mathcal{I}_{\text{cal}}|)$-th empirical quantile of $\{S_i^{\text{CQR-m}}\}_{i \in \mathcal{I}_{\text{cal}}} \cup \{\infty\}$, with:

$$S_i^{\text{CQR-m}} = \max\left(\frac{\widehat{q}_{\frac{\alpha}{2}}(x_i) - y_i}{\widehat{q}_{\frac{1}{2}}(x_i) - \widehat{q}_{\frac{\alpha}{2}}(x_i)}, \frac{y_i - \widehat{q}_{1-\frac{\alpha}{2}}(x_i)}{\widehat{q}_{1-\frac{\alpha}{2}}(x_i) - \widehat{q}_{\frac{1}{2}}(x_i)}\right). \tag{25}$$

In addition, there is an improved version of CQR-m, which does not require estimating the quantile of the regression median Sesia & Candès (2020). The prediction interval it constructs is as follows:

$$\mathcal{C}^{\text{CQR-r}}(x_{n+1}) = \left[\widehat{q}_{\frac{\alpha}{2}}(x_{n+1}) - \widehat{\Delta}_{\alpha}^{\text{CQR-r}}, \widehat{q}_{1-\frac{\alpha}{2}}(x_{n+1}) + \widehat{\Delta}_{\alpha}^{\text{CQR-r}}\right],$$
$$\widehat{\Delta}_{\alpha}^{\text{CQR-r}} = t_{1-\alpha}^{\text{CQR-r}}\left[\widehat{q}_{1-\frac{\alpha}{2}}(x_i) - \widehat{q}_{\frac{\alpha}{2}}(x_i)\right], \tag{26}$$

where $t_{1-\alpha}^{\text{CQR-r}}$ is the $(1-\alpha)(1+1/|\mathcal{I}_{\text{cal}}|)$-th empirical quantile of $\{S_i^{\text{CQR-r}}\}_{i \in \mathcal{I}_{\text{cal}}} \cup \{\infty\}$, with:

$$S_i^{\text{CQR-r}} = \max\left(\frac{\widehat{q}_{\frac{\alpha}{2}}(x_i) - y_i}{\widehat{q}_{1-\frac{\alpha}{2}}(x_i) - \widehat{q}_{\frac{\alpha}{2}}(x_i)}, \frac{y_i - \widehat{q}_{1-\frac{\alpha}{2}}(x_i)}{\widehat{q}_{1-\frac{\alpha}{2}}(x_i) - \widehat{q}_{\frac{\alpha}{2}}(x_i)}\right). \tag{27}$$

### A.4.3 Conformal Quantile Regression with Full Model.

CQRFM builds upon CQR-m by introducing a modification that allows the model to output three distinct values: the lower bound, median, and upper bound simultaneously from a single neural network. The key idea is to train a neural network $\mathcal{N} : \mathbb{R}^d \to \mathbb{R}^3$ such that $\mathcal{N}(x) = (l(x), m(x), u(x))$, where $l$, $m$, and $u$ are functions that estimate the $\alpha/2$-quantile, the median, and the $(1 - \alpha/2)$-quantile, respectively, with the constraint that $l(x) \leq m(x) \leq u(x)$ for all $x \in \mathbb{R}^d$.

The network is trained using a modified quantile regression loss function:

$$L_\tau(\mathcal{N}(x), y) = h_{\tau/2}(y - l(x)) + h_{1/2}(y - m(x)) + h_{1-\tau/2}(y - u(x)), \tag{28}$$

where $h_\tau(u) = (\tau - \mathbf{1}_{u \leq 0})u$ is the standard quantile regression loss function.

Similar to CQR-m, the prediction interval is constructed as:

$$\mathcal{C}^{\text{CQRFM}}(x_{n+1}) = \left[l(x_{n+1}) - \widehat{\Delta}_{\alpha,\text{lo}}^{\text{CQRFM}}, u(x_{n+1}) + \widehat{\Delta}_{\alpha,\text{up}}^{\text{CQRFM}}\right],$$
$$\widehat{\Delta}_{\alpha,\text{lo}}^{\text{CQRFM}} = t_{1-\alpha}^{\text{CQRFM}}\left[m(x_i) - l(x_i)\right], \tag{29}$$
$$\widehat{\Delta}_{\alpha,\text{up}}^{\text{CQRFM}} = t_{1-\alpha}^{\text{CQRFM}}\left[u(x_i) - m(x_i)\right],$$

where $t_{1-\alpha}^{\text{CQRFM}}$ is the $(1-\alpha)(1+1/|\mathcal{I}_{\text{cal}}|)$-th empirical quantile of $\{S_i^{\text{CQRFM}}\}_{i \in \mathcal{I}_{\text{cal}}} \cup \{\infty\}$, with:

$$S_i^{\text{CQRFM}} = \max\left(\frac{l(x_i) - y_i}{m(x_i) - l(x_i)}, \frac{y_i - u(x_i)}{u(x_i) - m(x_i)}\right). \tag{30}$$

### A.5 Mixture Distribution with 1D Input

Following Luo & Zhou (2025), we generate $n = 10000$ samples where predictors $X_i$ are drawn independently from $\text{Uniform}(0, 1)$. The response variables are sampled i.i.d. according to:

$$y \sim \text{Triangular}(0, x, x), \tag{31}$$

with conditional density:

$$f(y|x) = \frac{2y}{x^2}\mathbb{1}\{y \in (0, x)\}. \tag{32}$$

### A.6 Mixture Distribution with 2D Input

We generate $n = 10000$ samples where 2D predictors $X_i = (X_{i1}, X_{i2})$ are drawn independently from $\text{Uniform}(-1, 1)^2$. The response variable $Y_i$ is then sampled according to the quadrant-specific distributions:

$$Y_i|X_i \sim p(y|x_{i1}, x_{i2}), \tag{33}$$

where the conditional distribution $p(y|x_{i1}, x_{i2})$ is defined as:

$$p(y|x_{i1}, x_{i2}) = \begin{cases} \text{Uniform}(0, 1), & \text{if } x_{i1} \geq 0, x_{i2} \geq 0 \\ \text{Normal}(0, (\frac{1}{5})^2), & \text{if } x_{i1} < 0, x_{i2} \geq 0 \\ \text{Exponential}(\frac{1}{2}), & \text{if } x_{i1} < 0, x_{i2} < 0 \\ \frac{1}{2} \cdot \text{Normal}(-\frac{1}{2}, (\frac{1}{10})^2) + \frac{1}{2} \cdot \text{Normal}(\frac{1}{2}, (\frac{1}{10})^2). & \text{if } x_{i1} \geq 0, x_{i2} < 0 \end{cases} \tag{34}$$

We have also visualized the simulation experiment results (synthetic[2]) for multiple features. We provide two perspectives: a front view and a back view. It is evident from both views that CMR occupies a smaller proportion of the entire space. In this dataset, CHR exhibits lower tail coverage (as indicated by fewer green stars), while CMR demonstrates even lower tail coverage compared to CQR and CQRFM.

### A.7 Boxplots

In this section, we present box plots comparing various conformal regression methods across 14 datasets, evaluating their performance in terms of coverage, size, and tail coverage. For each dataset, each conformal regression method underwent 50 randomized partitioning tests with stratified training/test splits to ensure robustness of results.

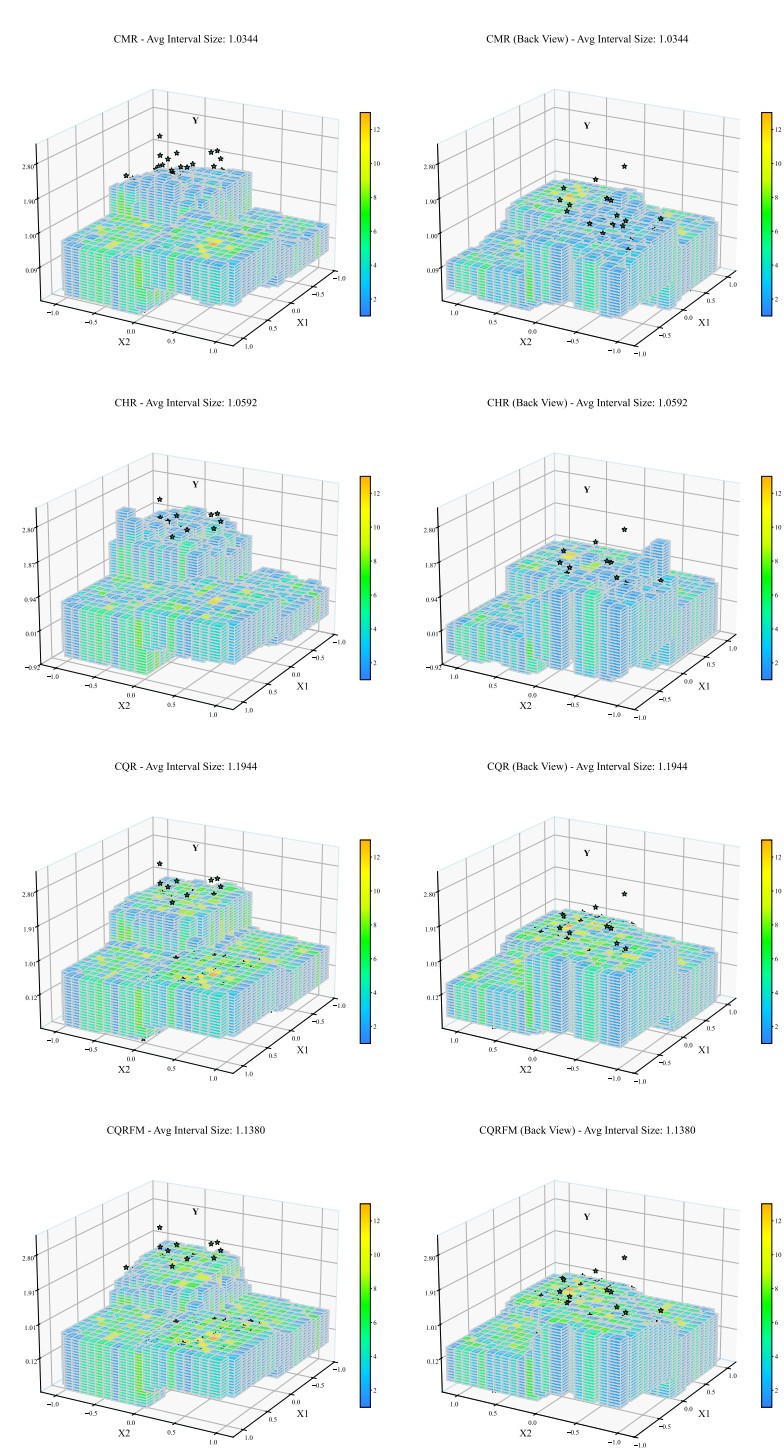

Figure 3: In this figure, we present the three-dimensional prediction intervals and coverage rates for CMR, CHR, CQR, and CQRFM. Similar to Figure 2, we use regional density to reflect the size of the intervals for different methods. Additionally, we provide the tail coverage performance of the models, where the green stars represent the points that are not covered. We offer two perspectives: a front view and a back view. From both views, it can be observed that the region occupied by CMR is smaller compared to the other methods.

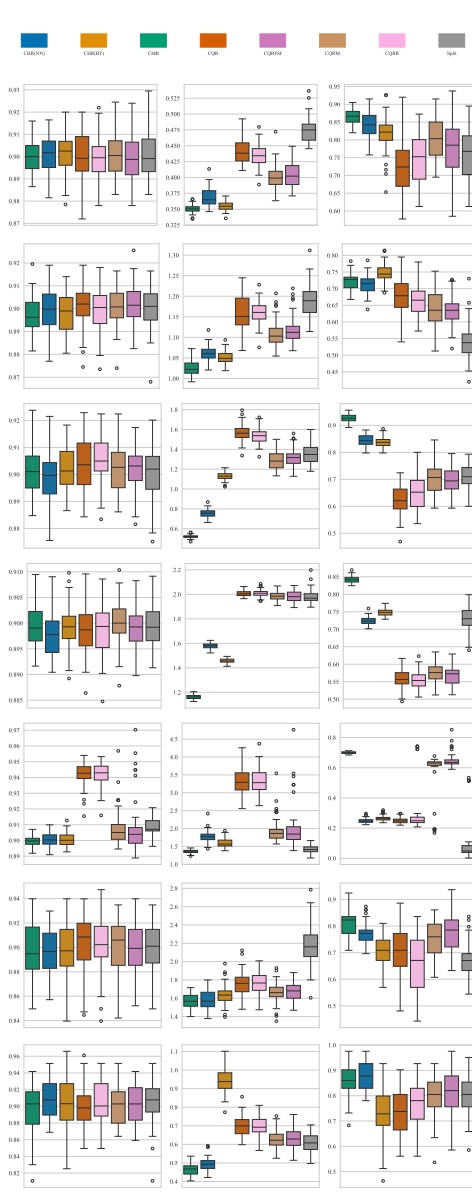

Figure 4: Box plots comparing various conformal regression methods across multiple datasets and evaluation metrics. Each experiment was conducted with 50 random dataset splits. The datasets are arranged vertically from top to bottom: "synthetic[1]", "synthetic[2]", "bike", "bio", "blog", "community", and "concrete". The evaluation metrics, displayed horizontally from left to right, include Coverage, Size, and TCR.

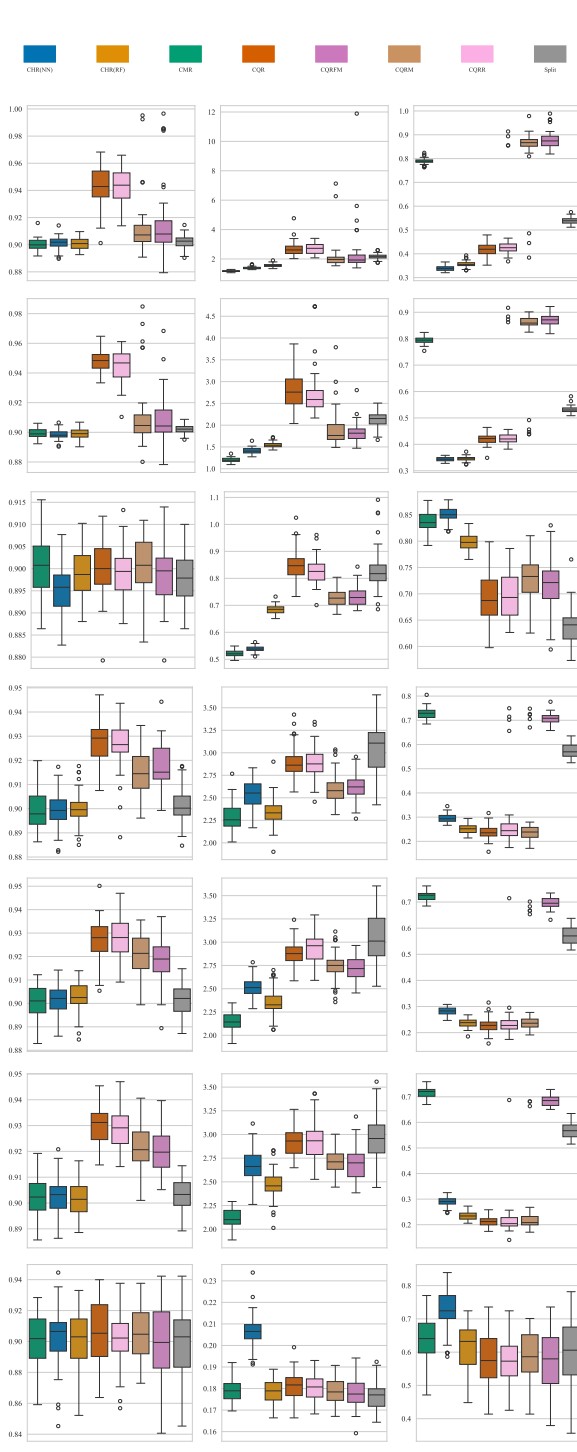

Figure 5: Box plots comparing various conformal regression methods across multiple datasets and evaluation metrics. Each experiment was conducted with 50 random dataset splits. The datasets are arranged vertically from top to bottom: "facebook1", "facebook2", "homes", "meps19", "meps20", "meps21", and "star" . The evaluation metrics, displayed horizontally from left to right, include Coverage, Size, and TCR.

A.8 SENSITIVITY ANALYSIS

For synthetic[1] experiments, each parameter configuration was randomly partitioned and evaluated across 10 independent trials, with results averaged to ensure statistical robustness. The hyperparameter grid included $m = [100, 200, 500, 1000, 2000, 5000]$ and $\sigma = [0.001, 0.01, 0.1, 0.5]$. Dataset partitions followed a 5600-sample training set, 2400-sample calibration set, and 2000-sample test set configuration. All experimental protocols (including supplementary experiments) were executed using an NVIDIA GeForce RTX 4060 GPU with 16GB memory.

We denote the best metric in bold for cases with the same $m$ but different $\sigma$ values, and underline the best metric for cases with the same $\sigma$ but different $m$ values. As can be observed from the table, the model performs optimally when $\sigma$ is set to 0.01. As $m$ increases, which corresponds to an increase in the number of generated predictive samples, the size of the prediction set decreases while the tail coverage rate increases. However, this also incurs a certain computational cost in terms of calibration time. Additionally, even when $m$ is set to 100, the model can still maintain sufficient coverage and a relatively small prediction set size.

Table 3: Performance Comparison: CMR Model Performance Under Different m and $\sigma$ Values

| Dataset | m | $\sigma$ | Coverage (%) | Interval Size | Tail Coverage (%) | Train Time (s) | Calib Time (s) | Test Time (s) |
|---|---|---|---|---|---|---|---|---|
| synthetic[1] | 100 | 0.001 | 89.825 (1.070) | 0.362 (0.008) | 85.050 (1.920) | 59.704 | 83.568 | 74.288 |
| synthetic[1] | 100 | 0.010 | 89.650 (1.282) | **0.358 (0.008)** | 85.575 (1.743) | 58.474 | 82.629 | 74.781 |
| synthetic[1] | 100 | 0.100 | 89.985 (0.938) | 0.385 (0.008) | **91.150 (1.463)** | 59.371 | 82.578 | 75.118 |
| synthetic[1] | 100 | 0.500 | 89.790 (1.005) | 0.594 (0.004) | 85.825 (2.264) | 59.265 | 82.949 | 74.765 |
| synthetic | 200 | 0.001 | 89.650 (1.046) | 0.354 (0.005) | 85.600 (1.586) | 59.560 | 83.027 | 74.687 |
| synthetic[1] | 200 | 0.010 | 89.610 (0.867) | **0.352 (0.007)** | 86.025 (1.429) | 59.006 | 82.983 | 77.334 |
| synthetic[1] | 200 | 0.100 | 89.840 (0.691) | 0.378 (0.004) | **92.175 (1.401)** | 74.496 | 103.748 | 96.643 |
| synthetic[1] | 200 | 0.500 | 89.350 (0.919) | 0.545 (0.008) | 84.150 (2.762) | 73.409 | 103.822 | 94.213 |
| synthetic[1] | 500 | 0.001 | 89.665 (1.057) | 0.350 (0.004) | 85.825 (1.323) | 73.252 | 104.748 | 94.268 |
| synthetic[1] | 500 | 0.010 | 89.575 (1.081) | **0.347 (0.006)** | 86.450 (1.308) | 73.261 | 105.462 | 94.479 |
| synthetic[1] | 500 | 0.100 | 89.885 (0.993) | 0.374 (0.007) | **93.375 (1.398)** | 73.586 | 104.831 | 94.194 |
| synthetic[1] | 500 | 0.500 | 89.215 (1.002) | 0.503 (0.008) | 83.325 (2.620) | 72.968 | 107.867 | 94.514 |
| synthetic[1] | 1000 | 0.001 | 89.775 (0.879) | 0.348 (0.005) | 86.175 (1.173) | 70.581 | 111.678 | 97.950 |
| synthetic[1] | 1000 | 0.010 | 89.850 (1.000) | **0.346 (0.005)** | 86.975 (1.325) | 69.100 | 111.217 | 96.935 |
| synthetic[1] | 1000 | 0.100 | 89.930 (1.057) | 0.373 (0.006) | **94.075 (1.475)** | 64.220 | 94.174 | 79.935 |
| synthetic[1] | 1000 | 0.500 | 89.690 (1.141) | 0.485 (0.009) | 83.475 (2.916) | 66.124 | 98.262 | 85.317 |
| synthetic[1] | 2000 | 0.001 | 89.750 (1.032) | 0.347 (0.006) | 86.700 (1.208) | 66.229 | 100.781 | 83.691 |
| synthetic[1] | 2000 | 0.010 | 89.845 (1.009) | **0.344 (0.005)** | 87.350 (1.361) | 57.984 | 88.392 | 74.747 |
| synthetic[1] | 2000 | 0.100 | 89.900 (1.070) | 0.371 (0.007) | **94.400 (1.184)** | 49.222 | 78.070 | 65.324 |
| synthetic[1] | 2000 | 0.500 | 89.585 (1.101) | 0.468 (0.009) | 82.125 (3.509) | 50.378 | 80.410 | 67.836 |
| synthetic[1] | 5000 | 0.001 | 89.865 (0.903) | 0.346 (0.006) | 86.700 (0.843) | 70.462 | 116.747 | 87.807 |
| synthetic[1] | 5000 | 0.010 | 89.820 (1.044) | **0.343 (0.005)** | 87.300 (1.150) | 64.484 | 106.798 | 76.157 |
| synthetic[1] | 5000 | 0.100 | 89.850 (0.956) | 0.370 (0.005) | **94.775 (1.186)** | 57.479 | 108.471 | 79.874 |
| synthetic[1] | 5000 | 0.500 | 89.740 (0.981) | 0.458 (0.007) | 82.025 (3.552) | 60.124 | 123.556 | 81.857 |

A.9 COMPARISON OF TIME CONSUMPTION OF DIFFERENT METHODS

The inference process can be broken down into two stages:

1. **Base Model Sampling:** Generating $m$ samples for a given test point using the trained CFM model, RF and NN. This cost is a fundamental overhead shared by all CFM - based conformal methods we compared (CHR, CQR, and ours).

2. **Conformal Interval Generation:** Utilizing these samples to compute the final prediction interval. The cost difference in this stage reflects the intrinsic complexity of each conformal algorithm.

To quantify these costs, we conducted an ablation study using CFM as the fixed base model and measured the time taken by different conformal algorithms. The results are as follows:

A.10 PART 1: CONDITIONAL FLOW MATCHING MODEL

This section defines the core generative model, which learns the conditional distribution $p(y|x)$ using a simulation-free objective.

| Method | Train Time | Calibration Sampling Time | $\hat{q}$ Calculation Time | Test Sampling Time | Prediction Interval Generation Time | Generation time of prediction interval for a single sample |
|--------|------------|---------------------------|----------------------------|--------------------|-------------------------------------|-----------------------------------------------------------|
| CHR (CFM) | 64.220s | 1.368s | 70.230s | 1.104s | 0.107s | 0.606ms |
| CQR (CFM) | 64.325s | 0.207s | 1.245s | 0.925s | 0.030s | 0.478ms |
| CMR (CFM) | 64.224s | 1.346s | 92.264s | 0.988s | 78.560s | 39.774ms |
| CMR (RF) | 10.781s | 0.073s | 0.063s | 0.059s | 0.005s | 0.003ms |
| CMR (NN) | 120.854s | 0.003s | 0.068s | 0.003s | 0.005s | 0.003ms |
| CHR (RF) | 10.792s | 0.049s | 79.102s | 0.048s | 66.006s | 33.003ms |
| CHR (NN) | 120.894s | 0.002S | 80.825s | 0.003s | 66.332s | 33.166ms |
| CQR | 12.183S | 0.002S | 0.046s | 0.003s | 5.183s | 2.592ms |

Table 4: A comparison of the time consumption of different methods. In the simulation experiment, each method was tested 10 times.

```python
import torch
import torch.nn as nn
import numpy as np

class FlowMatchingModel(nn.Module):
    """Defines the neural network and the training/sampling logic for the CFM model
    ."""

    def __init__(self, context_dim, hidden_dim=64, num_layers=3, sigma=0.01):
        super().__init__()
        self.sigma = sigma

        # Neural network to approximate the vector field v_theta(y, t, x)
        layers = [nn.Linear(1 + 1 + context_dim, hidden_dim), nn.SiLU()]
        for _ in range(num_layers - 1):
            layers.extend([nn.Linear(hidden_dim, hidden_dim), nn.SiLU()])
        layers.append(nn.Linear(hidden_dim, 1))
        self.net = nn.Sequential(*layers)

    def forward(self, y, t, context):
        net_input = torch.cat([y, t.reshape(-1, 1), context], dim=1)
        return self.net(net_input)

    def train_model(self, train_loader, epochs=50, lr=1e-3, device='cpu'):
        self.to(device)
        optimizer = torch.optim.Adam(self.parameters(), lr=lr)

        self.train()
        for epoch in range(epochs):
            for x_batch, y_batch in train_loader:
                x_batch, y_batch = x_batch.to(device), y_batch.to(device)
                optimizer.zero_grad()

                t = torch.rand(x_batch.size(0), 1, device=device)
                z0 = torch.randn_like(y_batch)

                yt = t * y_batch + (1 - t) * z0 + self.sigma * torch.randn_like(
    y_batch)
                ut = y_batch - z0 # Target vector field

                vt_pred = self(yt, t, x_batch)
                loss = nn.functional.mse_loss(vt_pred, ut)

                loss.backward()
                optimizer.step()

    def sample(self, x_cond, num_samples, steps=100):
        # Generate samples by solving the ODE from t=0 to t=1
        self.eval()
        with torch.no_grad():
            y_t = torch.randn(num_samples, 1, device=next(self.parameters()).device
    )
            dt = 1.0 / steps
            for t_val in np.linspace(0, 1 - dt, steps):
                t = torch.full((num_samples, 1), t_val, device=y_t.device)
                context = x_cond.repeat(num_samples, 1)
                dy = self(y_t, t, context) * dt
                y_t = y_t + dy
        return y_t.cpu().numpy().flatten()
```

Listing 1: Flow Matching Model: The Generative Core

## A.11  PART 2: CONFORMAL REGRESSION ALGORITHM

This section implements the conformalization procedure. It uses a trained `Flow Matching Model` to perform calibration and construct prediction intervals.

```python
class CMR:
    """Implements the calibration and prediction logic for Conformal Regression."""

    def __init__(self, model):
        self.model = model
        self.q_threshold = None # Calibrated quantile

    def calibrate(self, cal_loader, alpha, m_samples=1000):
        # Phase 2: Calibration to find the threshold q
        scores = []
        device = next(self.model.parameters()).device
        for x_cal, y_cal in cal_loader:
            for i in range(x_cal.size(0)):
                xi, yi = x_cal[i:i+1].to(device), y_cal[i].item()

                y_samples = self.model.sample(xi, m_samples)
                y_samples.sort()

                # Nonconformity score: min number of points in the shortest
                interval covering yi
                score = self._find_min_points_to_cover(y_samples, yi)
                scores.append(score)

        n_cal = len(scores)
        q_level = np.ceil((1 - alpha) * (n_cal + 1)) / n_cal
        self.q_threshold = int(np.quantile(scores, q_level, method="higher"))

    def predict(self, x_test, m_samples=1000):
        # Phase 3: Prediction Interval Construction
        if self.q_threshold is None:
            raise RuntimeError("CMR must be calibrated first. Call .calibrate()")

        device = next(self.model.parameters()).device
        y_samples = self.model.sample(x_test.to(device), m_samples)
        y_samples.sort()

        # Find the shortest interval containing q_threshold points
        interval = self._find_shortest_interval(y_samples, self.q_threshold)
        return interval

    def _find_min_points_to_cover(self, sorted_samples, y_true):
        # Find the smallest k such that the shortest interval of k points covers
        y_true
        for k in range(2, len(sorted_samples) + 1):
            shortest_interval = self._find_shortest_interval(sorted_samples, k)
            if shortest_interval[0] <= y_true <= shortest_interval[1]:
                return k
        return len(sorted_samples)

    def _find_shortest_interval(self, sorted_samples, k_points):
        # Find the shortest interval containing exactly k_points using a sliding
        window
        k_points = min(k_points, len(sorted_samples))
        min_width = float('inf')
        best_interval = (sorted_samples[0], sorted_samples[-1])

        for i in range(len(sorted_samples) - k_points + 1):
            width = sorted_samples[i + k_points - 1] - sorted_samples[i]
            if width < min_width:
                min_width = width
                best_interval = (sorted_samples[i], sorted_samples[i + k_points -
1])
        return best_interval
```

Listing 2: CMR: The Conformal Prediction Algorithm

