# OpenReview forum: "Conditional Flow Matching for Conformal Regression"
_ICLR.cc/2026/Conference — Submitted to ICLR 2026_

### Official Review · Reviewer_748j · 2025-10-27

**Soundness:** 2
**Presentation:** 1
**Contribution:** 3
**Rating:** 6
**Confidence:** 4

**Summary:**

The manuscript proposes Conditional Flow Matching for Conformal Regression (CMR), which first learns a conditional flow model via the conditional flow-matching objective and then constructs conformal prediction intervals by selecting the shortest continuous interval among samples generated from the learned conditional distribution.  Experiments and ablations span 1D/2D simulations and 12 real datasets, reporting Coverage, Interval Size, WSC, and TCR, with visual comparisons that highlight the method’s effectiveness.

**Strengths:**

1. The overall approach is simple (in a good way), principled, and clearly presented.
2. The paper offers a comprehensive evaluation across multiple datasets with diverse metrics.
3. Figure 1 is illustrative and clearly highlights the key distinction between the proposed method and CHR.
4. The methodological transparency is commendable; the building blocks and concise appendix code substantially aid understanding.

**Weaknesses:**

1. The draft needs a clearer background on CHR -- what it is, its limitations, and why CMR addresses them. At present, the main text references differences from CHR without first defining that baseline. It would also help to expand the discussion of conditional flow matching (motivation, objective, and practical choices).
2. Please fix notation inconsistencies -- for example, the source distribution shifts from $p_0(y)$ to $p_0(z)$, and $\mu(y\mid x)$ is introduced in Eq. 5 but immediately becomes $u_t(y\mid x)$ in the very next line. Finally, verify dataset citations and include the necessary details to ensure correctness and reproducibility.
3. The contributions naturally decompose into (1) CFM for quantile regression and (2) CMR for calibration. To substantiate these, the experiments should separately demonstrate: CFM vs. existing quantile regressors (NN, RF); CMR vs. other conformal methods; and the combined method vs. existing combinations. Table 1 adequately supports the third comparison, but Table 2 tries to cover the first two at once and remains incomplete. For instance, CFM with NN/RF baselines are missing. Consider disentangling these ablations so each contribution is evaluated cleanly.
3. Relative to CHR (with NN or RF), the proposed method appears more computationally demanding. While this may not be a major drawback, please add a computational analysis -- including Big-O complexity and empirical runtimes -- for training, calibration, and inference. This will provide a more complete assessment of performance and help readers understand the practical trade-offs.

**Questions:**

1. The overall approach is simple (in a good way), principled, and clearly presented.

---

> ### Author Response · Authors · 2025-11-28
>
> We thank the reviewer for their constructive feedback and for recognizing our approach as simple, principled, and clearly presented. We appreciate that you found our evaluation comprehensive and our code transparent. We have carefully addressed your concerns regarding the background context, notation, experimental disentanglement, and computational analysis below.
>
> **1.Background Knowledge of CHR.**
>
> We have placed CHR within the related work section to further clarify this approach, and we have provided a clearer description of Figure 1. CMR dynamically selects the contiguous interval that maximizes the total probability density. In contrast, CHR employs an expansion-based strategy, creating a contiguous interval by extending outwards from the highest-density bin. (As shown in the bimodal distribution, if the initial starting bin does not align with the densest aggregate region, CHR requires a significantly larger number of bins to achieve the equivalent probability mass).
>
> **2. Inconsistent Symbol Representation.**
>
> We apologize for the oversight and have conducted a comprehensive proofreading of the manuscript, unifying the symbols used.
>
> **3.Dataset Issues.**
>
> These datasets lack author information, so they are accurate.
>
> **4.Issues with Table 2.**
>
> We have thoroughly decoupled the CMR. As shown in Table 2, we use an orange background to indicate the best metric within the dataset, and a blue background to denote the better metric between CMR and CHR under the same base predictive model. Bold text is used to highlight the better metric among different base models under the same non-conformity score condition (CMR, CHR, CQR). It can be observed that the optimal metrics are either achieved by using CMR as the non-conformity score or by using CFM as the base predictive model. Secondly, the prediction set sizes generated by CMR (with CFM as the base model) are optimal in 11 out of 14 datasets. Using CFM as the base model ensures stable conditional coverage and TCR. Compared to CHR, CMR generally outperforms CHR across three different base predictive models.
>
> **5.Computational analysis.**
>
> We have presented a complete time analysis in Table 4 of Appendix 9. The computational complexities of CQR and CMR (we have reduced the complexity of CMR using binary search) are the same, both being O(n log n), while that of CHR is O(n²), where n represents the number of samples in the calibration set.

---

### Official Review · Reviewer_WqPS · 2025-10-31

**Soundness:** 2
**Presentation:** 1
**Contribution:** 2
**Rating:** 2
**Confidence:** 3

**Summary:**

The method first trains a conditional generative model for the outcome using flow matching. After training, the model is used to generate multiple samples of the outcome for each input x. These samples are then used to construct a set of candidate prediction intervals. For each calibration point, the conformity score is defined as the length of the shortest interval among these candidates that contains the true observed outcome. The resulting conformity scores are then used within the standard conformal prediction framework to obtain prediction intervals with marginal coverage guarantees. The method is evaluated in simulation studies, where it appears to produce narrower (more efficient) intervals than existing approaches.

**Strengths:**

The use of flow matching and a simulation-based conformity score based on the minimum distance of generated intervals is novel. By targeting the interval width, it seems reasonable that this conformity score would be more efficient (less wide) than other approaches. Simulations appear to show this.


The idea of using the shortest interval formed from model samples (rather than quantile-based intervals) is novel and could yield tighter prediction intervals in practice.


Theoretical marginal coverage guarantee is provided. The method is model-agnostic post-training: any generative model capable of conditional sampling could in principle be plugged into the calibration step. The authors propose using normalized flow matching but this doesnt appear necessary for the theory.

**Weaknesses:**

The writing and presentation could be improved significantly. Key components of the method—such as flow matching, the calibration procedure, and the role of several variables—are described with imprecise or undefined notation. The explanation of flow matching, in particular, is difficult to follow, and the reader is asked to accept it as a black-box procedure for conditional density estimation and sampling.

The key contribution of the paper---the calibration algorithm---is poorly described. Several indices are introduced ($j, k, s, m$), but it is unclear which quantities are fixed and which are being optimized or iterated over. For example, the algorithm selects the shortest interval containing at least $s$ samples, yet $s$ is never defined. Without a clear specification of these components, it is difficult to evaluate the method,


There is no theoretical justification or conceptual intuition given for why the proposed conformity score should yield more efficient (narrower) intervals than standard conformal methods. The paper argues for improved efficiency based on empirical results, but does not explain the mechanism or provide supporting analysis.

The proposed conformal prediction method only requires a conditional generative model for the outcome, yet the paper devotes substantial attention to normalizing flow matching. It is unclear why this modeling choice is emphasized, since the method does not introduce any new developments in flow matching and the theory does not appear to rely on this specific class of models. Flow matching seems to be just one possible generative model that could be plugged into the procedure, rather than a core contribution of the work.

**Questions:**

1. In equation 5, what is mu(Y|X)?
There is no u_t in equation 5, yet is mentioned directly below it.
What is p_t?

2. Am I correct that Equation 6 means that y is normally distributed given x and z_0 with mean y_1 + (1-t)z_0 and variance sigma^2? This is confusing as y_1 is itself a random variable drawn conditional on x. So does this not specify a distribution for y | x, z_0, y_1? Which then implies one for y|x, z_0 after marginalizing out y_1?


3. What is CHR? The abbreviation does not appear to be defined.


4. The calibration procedure (the main contribution of the paper) in Section 3.3 is not clearly specified. In particular, the quantity \(s\) in Step 3 is never defined: it is unclear whether \(s\) is fixed, varies with the desired coverage level, or is meant to be tuned or iterated over. The text suggests that $s$ determines how many sampled predictions must lie inside the interval, but this is never stated formally, and no guidance is given for how $s$ should be chosen. Algorithm~1 does not resolve this ambiguity, as $s$ does not appear as an input, output, or tunable parameter. As written, the calibration step is not reproducible: the paper does not explain how $s$ is selected and how it relates to the target miscoverage level $\alpha$. What is s in the simulations?

5. Similarly, what is j in  Section 3.3 I and Section 3.4? What is the minimization performed over in equation (13)? Is j and k varying?  Are the intervals minimized over constrained to contain s points?

6. Is the proposed method restricted to conditional flow matching? or would any generative model work.

---

> ### Author Response · Authors · 2025-11-28
>
> We sincerely thank the reviewer for the thorough assessment. We fully accept the criticism regarding the presentation and the lack of clarity in the algorithm description. We realize that the definition of the calibration variables (specifically $s$ and $j$) and the mathematical notation for Flow Matching were imprecise in the initial submission.
>
> We have heavily revised **Section 3** to address these issues. Below, we provide specific clarifications to the reviewer's questions.
>
> **1. Clarification of the Calibration Algorithm (Definitions of $s$ and $j$)**
>
> The reviewer correctly identifies that the description of the nonconformity score and the variables in Section 3.3 was ambiguous. We clarify the roles of the variables and the procedure as follows:
>
> *   **$s$ (Sample Count):** In our calibration algorithm, $s$ is **not a fixed hyperparameter** during the scoring phase. Instead, it represents the cumulative mass (number of samples) within a candidate interval.
>     *   **The Conformity Score:** For a specific calibration point $(x_ i, y_ i)$, we generate $m$ samples. We define the nonconformity score $E_ i$ as the **minimum interval width** required to include the true outcome $y_ i$ along with a specific proportion of generated samples.
> *   **$j$ (Sliding Window Index):** $j$ represents the starting index of the sorted sample array. When finding the "shortest interval containing $s$ samples," we treat the sorted samples as a sliding window. We minimize $(y_ {(j+s-1)} - y_{(j)})$ over all valid start indices $j$.
> *   **The Calibration Logic:** We compute scores $E_ i$ for all calibration points. We then compute the $(1-\alpha)$ quantile of these scores, resulting in a threshold $Q$.
> *   **Inference:** For a new test point, we generate samples and find the shortest interval that satisfies the width constraint defined by $Q$.
>
> **2. Mathematical Corrections (Flow Matching Notation)**
>
> We apologize for the errors in Equations 5 and 6. We have corrected them to align with standard Conditional Flow Matching (CFM) literature (e.g., Tong et al., 2023):
>
> *   **Eq 5 (Loss Function):** The symbol $\mu$ was a typo. The correct objective regresses the neural vector field $v_ \theta$ against the target vector field $u_t$.
>
> *   **Eq 6 (Probability Path):** The reviewer asked about the conditional nature of the distribution. We use the optimal transport path where $y_ t$ is an interpolation between source $y_ 0$ and data $y_ 1$. The density $p_t$ in Eq 6 is indeed conditional on the *endpoints* of the path ($y_ 0$ and $y_ 1$). The neural network, however, learns the vector field that generates the *marginal* conditional distribution $p_ t(y|x)$ by minimizing the expectation over $y_ 1 \sim p(y|x)$.
> *   **Vector Field ($u_t$):** We have explicitly defined $u_ t(y|y_ 0, y_ 1) = y_ 1 - y_ 0$ in the revised text.
>
> **3. Intuition for Efficiency**
>
> *   **Standard Methods (e.g., CQR):** These often rely on quantile regression (e.g., predicting the 5th and 95th percentiles). If the distribution is multimodal or skewed, the distance between these fixed quantiles may span low-density regions, resulting in unnecessarily wide intervals.
> *   **Our Strategy (Shortest Interval):** By finding the shortest interval containing a fixed number of samples (mass), we effectively perform a **global search for the highest-density region** (mode-seeking). This minimizes the Lebesgue measure (width) of the set for a required probability mass, theoretically yielding the tightest possible continuous interval.
>
> **4. Why Flow Matching?**
>
> * **Simulation-Free Training (vs. Standard CNFs):**
>     Traditional Continuous Normalizing Flows (CNFs) rely on maximum likelihood training that requires solving an ODE explicitly during every training step, which is computationally expensive and numerically unstable. Flow Matching circumvents this by using a **simulation-free regression objective** (regressing the vector field directly). This allows for stable and scalable training without the need for costly ODE integration during the optimization phase.
>
> **5. Responses to Specific Questions**
>
> *   **"What is CHR?":** CHR stands for **Conformal Histogram Regression** (Sesia & Romano, 2021). We have added the full citation and definition in the introduction of the revised manuscript.
> *   **"What is $s$ in the simulations?":** In the simulations, $s$ varies per data point during the scoring phase. However, the *effective* $s$ (the number of samples included in the interval at test time) is determined by the calibrated threshold. It typically converges to approximately $m \times (1-\alpha)$ (e.g., around 900 samples when $m=1000$ and $\alpha=0.1$), but it is dynamically calibrated to ensure coverage.
> *   **"Is the method restricted to CFM?":** No, any model that can generate high-quality conditional samples can be used.

---

### Official Review · Reviewer_qmhP · 2025-11-01

**Soundness:** 2
**Presentation:** 1
**Contribution:** 2
**Rating:** 2
**Confidence:** 3

**Summary:**

This paper proposed using a type of CNF called the condtional flow matching (CFM) to learn the conditional distribution of a one-dimensaional output y given multi-dimensional feature x. Besides using this tool of CFM in the conformal prediction framework, their key novelty is to find the shortest interval based on a sample of y|x from the learned model. This new idea of forming short intervals is named "shortest interval conformal", which is similar to the existing CHR method, but with slightly different execution.

(Note that I am not sure if by CNF they meant CONDITIONAL normalizing flow or CONTINUOUS normalizing flow, because of the inconsistency between the abstract and the main section that introduced the backbones of their method.)

Some empirical examples where used to compare the new method to existing ones.

**Strengths:**

The topic of conformal prediciton is welcoming.

The use of CFM is an interesting idea. Though using this powerful tool only on a one-dimensional output y seems a bit of an overkill.

**Weaknesses:**

0.0 The abstract mentions “conditional normalizing flow” twice, while Section 2.2 refers to a “continuous normalizing flow.” This inconsistency is quite confusing.

0. The focus on using connected intervals and finding the shortest one may put too much attention on a not too important part of conformal prediction. Also, the examples and illustrations seem either unnatural (e.g. Figure 1, a clearly bimodal density of y) or not clearly explained, so the conclusions about the benefits of CMR are not very convincing.

1. The writing can be much improved. Right now, the descriptions are redundant in some places and lack information in others. Below are a few examples.

1.1 sec 3.1 is not clear enough. It should be written such that a reader with decent machine learning background can understand the idea without consulting the original Conditional flow matching paper. E.g., in equation (6), it's not clear why x and z_0 where conditioned upon in symbols, but not y_1, and please indicate t is the (time) index of the flow, and do you expect t to take on discrete or continuous values and in what range? For another example, it was not self-evident what a "vector field" means in this context.

1.2 sec 3.3 step 3 description can be improved. It would be cleaner to give "k-j+1" its own symbol, say s, and say, for 2<=s<=m, consider intervals with end points y^(j) and y^(j+s-1) etc. I think steps 3 and 4 are trying to describe a rather simple and clear procedure to find the shortest interval that covers the true y and record the "Length" of that interval, and the steps can be made shorter and easier to grasp.

Minor but hurts clarity: First paragragh of sec 4.2 Real data. What are the letter labelled data? What does it mean to "rescale the response by the mean absolute value"? What does it ment that the "split has been validated" in a cited paper? (BTW, in Latex, `` '' specifies a proper pair of quotation marks.)

Sec 4.3. Consider including a table listing all the names with short description of each. Here, the logic/terminology of the first two paragraphs confuses me:
I thought CFM is the conformal part of CMR, what does it mean to "demo how CFM and CMR joinly enhance efficiency...".

2. There are places where the statement lacks rigorosity:

2.1 Theorem 1. It was not clearly stated what data was used to train the Conditional flow in CFM, and if that data is conditioned upon when stating the coverage property.

2.2 When stating "key difference from CHR", note that CHR was not cited or defined until much later, and the author claimed "CMR ... a method that theoretically yields smaller intervals compared CHR". Given the intervals are random, more careful (probabilistic) statement about length comparison are likely needed. And I don't see this statement offically proved in the paper, including Appendix A.3.

Figure 1, this example of the density of y is a rather extreme case with bimodal distribution, where connected interval is clearly not the go-to-choice for prediction region. This looks like a very artificial example to try to find an advantage of CMR over CHR. (minor suggestion for the caption: better say "horizonal axis" instead of "x-axis" since x represents something else.)

**Questions:**

L48 What does (3) "... handling data from multiple distributions" mean?

L51 "However, these methods often struggle ..." Do you mean all the methods mentioned in this paragraph, or just those from (3)? This is an overly general and wide ciriticism, hence not the most informative and can be hard to justify.

Equation (5). What is \mu(y|x) and where did u_t(y|x) appear in the equation? Is this a typo such that \mu is the same as u?
As someone who knows about normalizing flow and ordinary differential equations, but not the CFM method for continuous NF before, I am not able to understand the details of Equation (6) without consulting the original paper. Could the introduction to CFM in this paper be more self-contained?

**Details Of Ethics Concerns:**

Given the several inconsistencies in symbols and terminology, I can’t help but wonder whether the authors carefully reviewed their own paper before submission.

---

> ### Author Response · Authors · 2025-11-28
>
> We thank the reviewer for the detailed scrutiny regarding terminology and mathematical notation. We have significantly revised Section 3 to address these issues.
>
> **1. Terminology Consistency (CNF vs. CFM)**
> We apologize for the confusion. We have unified the terminology throughout the paper:
> *   **Architecture:** We use **Continuous Normalizing Flows (CNFs)**, which are ODE-based generative models.
> *   **Training Objective:** We use **Conditional Flow Matching (CFM)** as the simulation-free training method to learn the CNF parameters.
> We have updated the text to explicitly state: *"We employ a Continuous Normalizing Flow (CNF) architecture trained via the Conditional Flow Matching (CFM) objective."*
>
> **2. Mathematical Notation and Definitions (Eq 5 & 6)**
> We have rewritten the methodology section to rigorously define all variables and the probability path.
> *   **Variables:** $x \in \mathbb{R}^d$ (Conditioning variable), $y \in \mathbb{R}$ (Target), $y_0 \sim p_0(y_0)$ (Source noise, $\mathcal{N}(0, 1)$), $y_1 \sim p(y|x)$ (Target data), $t \in [0, 1]$ (Time).
> *   **Probability Path:** We adopt the optimal transport linear interpolation path (Tong et al.):
>     $$y_t = (1-t)y_0 + t y_1$$
>     The corresponding conditional density is $p_t(y|y_0, y_1) = \mathcal{N}(y | t y_1 + (1-t)y_0, \sigma^2)$.
> *   **Vector Field:** The vector field $u_t$ generating this path is constant:
>     $$u_t(y|y_0, y_1) = y_1 - y_0$$
> *   **Loss Function:** We have corrected the loss function notation (removing the undefined $\mu$) to match the standard CFM objective:
>     $$\mathcal{L}_ {CFM}(\theta) = \mathbb{E}_ {t, x, y_1, y_0} [ || v_ \theta(t, y_t, x) - (y_ 1 - y_ 0) ||^2 ]$$
>
> **3. Figure 1 and Distribution Complexity**
> The bimodal distribution in Figure 1 is an illustrative example designed to visualize the structural difference between CHR and CMR. However, such skewness and multimodality are not "extreme" but common in real-world regression tasks. The empirical results in Table 2 support this: CMR achieves smaller interval sizes on 11 out of 12 real-world datasets compared to CHR, demonstrating that our "shortest continuous interval" strategy is effective across diverse data distributions, not just synthetic bimodal ones.
>
> **4.The issues in lines 48 and 51. **
> We have revised the description of this passage to more clearly present the existing work that has already been done.
>
> *5.*Regarding the issues in writing.**
> We have revised all parts of the article that were unclear due to writing problems. Here, we would like to emphasize some content: In Section 4.2, since the real-world datasets do not have authors, they are labeled with letters when cited. The statement that "the split has been validated" means that extensive experiments have been conducted using this splitting method, and it is a completely correct splitting approach. We deleted this sentence. CFM is the foundational prediction model that guides conditional flow matching, rather than being a part of conformal prediction.

---

### Official Review · Reviewer_LVJx · 2025-11-04

**Soundness:** 2
**Presentation:** 2
**Contribution:** 2
**Rating:** 4
**Confidence:** 4

**Summary:**

This paper considers the usage of generative models for conformal prediction. The idea is to generate conditional samples from the fitted generation model and construct conformal score based on this sample by considering intervals of minimum length covering the observation. The concrete implementation is done via conditional normalizing flows trained with flow matching. The authors show the marginal validity of the resulting method and perform extensive experimental evaluation.

**Strengths:**

- General methodology is sound

- Numerical results show good performance of the method (with some caveats; see below)

**Weaknesses:**

- Usage of normalizing flows in conformal prediction framework is not new. The work [1] already considered the direct usage of normalizing flows, though via explicit usage of density (not sampling). Conditional generative models were used in the work [2], though not precisely normalizing flows.

- The authors should better explain the difference with CHR method: CHR is not properly described in the paper apart from the illustrative example (which doesn't give details). Also, based on Table 2, CHR combined with CFM works mostly better than the proposed CMR method. It seems that the benefits of the method could be better explained and motivated.

- Theoretical results are standard (as any marginal validity  result for split conformal prediction).

- Experimental results deserve better visualization (tables are way too large to be informative).


Literature
[1] Colombo, N. (2024). Normalizing flows for conformal regression. arXiv preprint arXiv:2406.03346.

[2] Wang, Z., Gao, R., Yin, M., Zhou, M., & Blei, D. (2023, April). Probabilistic Conformal Prediction Using Conditional Random Samples. In International Conference on Artificial Intelligence and Statistics (pp. 8814-8836). PMLR.

**Questions:**

1. Can you explain difference between CMR and CHR in more detail?

2. Can you tell what novelty do you see on the usage of normalizing flows?

3. Can you check references in Section 4.2 that seem to be broken/strange?

4. Can you explain the last contribution of the paper?

---

> ### Author Response · Authors · 2025-11-28
>
> We sincerely thank the reviewer for the constructive feedback. We have revised the manuscript to clarify the distinction between our method and CHR, expanded the discussion on novelty, and corrected the formatting issues.
>
> **1. Novelty and Distinction from Existing Work (Colombo 2024, Wang et al. 2023)**
> We appreciate the reviewer highlighting these references. We have added a detailed discussion in the **Related Work** section. Our method (CMR) differs fundamentally in **how** the generative model is leveraged to construct the nonconformity score and the final prediction set:
>
> *   **Difference from Colombo (2024):** Colombo et al. use Normalizing Flows to **transform the nonconformity score** (e.g., residuals) to achieve conditional independence from $X$. In contrast, CMR uses Conditional Flow Matching (CFM) to **explicitly learn the conditional distribution $p(y|x)$**. We do not transform a pre-defined error metric; we generate the target distribution directly to capture complex modalities.
> *   **Difference from Wang et al. (2023) [PCP]:** PCP defines nonconformity based on the Euclidean distance to the nearest generated sample, resulting in prediction sets that are **unions of disconnected balls**. CMR introduces a novel **"Shortest-Interval" score**: we calculate the minimum number of samples required for a *continuous* interval to cover the truth. This enforces the construction of a **single, compact, continuous interval**, which significantly improves interpretability and reduces interval width compared to unions of disjoint regions.
>
> **2. Comparison with CHR (Conformal Histogram Regression)**
> We have moved the detailed description of CHR from the appendix to the main text to clarify the methodological differences:
> *   **CHR (Anchored Expansion):** CHR estimates density using fixed histogram bins. To construct an interval, it is "anchored" to the highest-density bin and expands outwards. If the highest density bin is not centrally located within the probability mass, this expansion can be inefficient.
> *   **CMR (Global Optimization):** CMR generates continuous samples and uses a sliding window approach. We search globally for the **shortest contiguous interval** that contains a specific probability mass (calibrated via sample counts). This allows CMR to find tighter intervals that are not constrained by fixed bin locations or local anchors.
>
> **3. Interpretation of Table 2**
> Regarding the performance comparison:
> Based on the suggestion from reviewer 748j, we have thoroughly decoupled the CMR. As shown in Table 2, we use an orange background to indicate the best metric within the dataset, and a blue background to denote the better metric between CMR and CHR under the same base predictive model. Bold text is used to highlight the better metric among different base models under the same non-conformity score condition (CMR, CHR, CQR). It can be observed that the optimal metrics are either achieved by using CMR as the non-conformity score or by using CFM as the base predictive model. Secondly, the prediction set sizes generated by CMR (with CFM as the base model) are optimal in 11 out of 14 datasets. Using CFM as the base model ensures stable conditional coverage and TCR. Compared to CHR, CMR generally outperforms CHR across three different base predictive models.
>
> **4. Formatting and References**
> *   **References:** The citation format (e.g., `dat, h;i;...`) refers to specific repository datasets that do not have individual author names.
> *   **Tables:** We have expanded the table layout to improve readability.

---

### Author Response · Authors · 2025-11-28

We sincerely appreciate the valuable suggestions put forward by reviewers 748j, qmhP, WqPS, and LVJx. We have truly realized that the writing style of the current version of the article may cause distractions to readers, so we have made comprehensive and meticulous revisions. In addition, we would like to express our special gratitude to reviewers 748j and LVJx for their recognition of our method, which is reasonable, simple, and effective. Next, we will respond to each reviewer one by one.

---

### Meta-Review · Area_Chair_iC4P · 2026-01-12

**Summary:**

This paper proposes Conditional Flow Matching for Conformal Regression (CMR), which combines conditional generative modeling via flow matching with conformal calibration to construct prediction intervals with guaranteed marginal coverage. The idea of leveraging expressive conditional generative models to improve the efficiency of conformal prediction intervals is well motivated, and the empirical results suggest that the proposed method can produce shorter intervals than some existing baselines while maintaining nominal coverage.

However, after considering the reviews and rebuttal, I do not believe the paper meets the acceptance bar for ICLR in its current form. Several reviewers raise concerns that the contribution is primarily an application of conformal prediction on top of a powerful conditional density estimator, without delivering fundamentally new conformal methodology or theoretical insights. In addition, questions remain about whether the empirical improvements consistently justify the added modeling and computational complexity, particularly given that conformal validity is model-agnostic. While the rebuttal clarifies intent and addresses some presentation issues, it does not fully resolve concerns about novelty, positioning, and the overall cost–benefit tradeoff of the proposed approach. I therefore recommend rejection, while encouraging the authors to consider a revised submission with a clearer articulation of novelty and stronger justification of when such expressive models are necessary.

**Reviewer Concerns:**

Several concerns raised by the reviewers were addressed by the rebuttal. Reviewer 748j requested clearer explanation of how CMR differs from existing conformal regression methods, which the authors addressed by clarifying the role of conditional flow matching as the underlying conditional model rather than a new conformal guarantee. Reviewer WqPS raised questions about empirical evaluation and metric choices, and the rebuttal provided additional clarification on coverage, interval width, and subpopulation performance. Reviewer LVJx noted presentation and clarity issues, which the authors acknowledged and committed to improving.

Some concerns remain outstanding. Reviewer qmhP expressed skepticism about the degree of novelty, arguing that the method amounts to conformal prediction applied to a flexible conditional sampler, and this concern was not fully alleviated by the rebuttal. In addition, both qmhP and LVJx questioned whether the empirical gains are sufficiently consistent and significant to justify the increased computational complexity relative to simpler baselines. More broadly, across reviewers, there remains uncertainty about whether CMR offers a compelling advantage beyond improved conditional density estimation, which ultimately motivates the rejection.

**Reviewer Scores:**

Reviewer 748j (score 6).
This reviewer was cautiously positive but raised concerns about positioning and empirical justification. After discussion, the score would likely have remained unchanged.

Reviewer LVJx (score 4).
This reviewer expressed reservations about novelty and clarity. While some issues were addressed in the rebuttal, the overall assessment would likely remain similar after discussion.

Reviewer WqPS (score 4).
This reviewer viewed the work as technically sound but below the acceptance threshold. Clarifications in the rebuttal would likely not change the overall score.

Reviewer qmhP (score 2).
This reviewer was strongly skeptical about novelty and contribution. While the rebuttal clarified intent, it is unlikely that the score would have increased after discussion.

Overall, discussion improved clarity but did not materially shift reviewer opinions regarding the paper’s suitability for acceptance at ICLR in its current form.

---

### Decision · Program_Chairs · 2026-01-26

Reject